# Structural insights into the activation mechanism of antimicrobial GBP1

Marius Weismehl [1,2], Xiaofeng Chu [3], Miriam Kutsch [4,5,6,7], Paul Lauterjung[8,9], Christian Herrmann [8], Misha Kudryashev [3,10] & Oliver Daumke [1,2]✉

## Abstract

**The dynamin-related human guanylate-binding protein 1 (GBP1) mediates host defenses against microbial pathogens. Upon GTP binding and hydrolysis, auto-inhibited GBP1 monomers dimerize and assemble into soluble and membrane-bound oligomers, which are crucial for innate immune responses. How higher-order GBP1 oligomers are built from dimers, and how assembly is coordinated with nucleotide-dependent conformational changes, has remained elusive. Here, we present cryo-electron microscopy-based structural data of soluble and membrane-bound GBP1 oligomers, which show that GBP1 assembles in an outstretched dimeric conformation. We identify a surface-exposed helix in the large GTPase domain that contributes to the oligomerization interface, and we probe its nucleotide- and dimerization-dependent movements that facilitate the formation of an antimicrobial protein coat on a gram-negative bacterial pathogen. Our results reveal a sophisticated activation mechanism for GBP1, in which nucleotide-dependent structural changes coordinate dimerization, oligomerization, and membrane binding to allow encapsulation of pathogens within an antimicrobial protein coat.**

**Keywords** Dynamin Superfamily; Electron Microscopy; Guanylate-Binding Proteins; GTPase; Oligomerization
**Subject Categories** Microbiology, Virology & Host Pathogen Interaction; Structural Biology

## Introduction

Guanylate-binding proteins (GBPs) are interferon-inducible, dynamin-related GTPases that mediate cell-autonomous immunity against a wide range of microbial pathogens (Kutsch and Coers, 2021; MacMicking, 2012; Meunier and Broz, 2016; Praefcke, 2018;

Santos and Broz, 2018). Host defense against intracellular bacterial pathogens such as *Shigella* and *Salmonella* is orchestrated by human GBP1 (GBP1), a cytosolic lipopolysaccharide (LPS) immune sensor and surfactant (Kutsch et al, 2020; Santos et al, 2020). Besides boosting the release of LPS from intracellular bacteria into the host cell cytosol (Goers et al, 2023) and accelerating activation of the non-canonical inflammasome caspase-4 to mediate pyroptosis of the infected cell (Dickinson et al, 2023; Fisch et al, 2019a; Santos et al, 2020; Wandel et al, 2020), GBP1 assembles on the surface of bacterial pathogens into an antimicrobial microcapsule or coatomer which breaks down the integrity of the bacterial envelope (Kutsch et al, 2020). Coatomer formation on intracellular gram-negative bacteria requires GBP1 to oligomerize into soluble polymers that bind to membrane-attached LPS and rearrange into a stable protein coat encapsulating the bacterial cell (Kutsch et al, 2020). Whereas targeting of bacteria by GBP1 is required to inhibit actin-based bacterial dissemination and facilitate antimicrobial-mediated bacterial lysis (Gaudet et al, 2021; Kutsch et al, 2020; Li et al, 2017; Piro et al, 2017; Wandel et al, 2017), coatomer formation is dispensable for acceleration of caspase-4 activation (Dickinson et al, 2023). Instead, pyroptosis is promoted by GBP1 and GBP2 polymers interacting with soluble LPS to form GBP-LPS hubs for non-canonical inflammasome activation (Dickinson et al, 2023).

Structurally, GBPs are composed of a dynamin-related large GTPase (LG) domain that features a unique "guanine cap" around the nucleotide-binding site, a helical middle domain (MD), and a helical GTPase effector domain (GED, Fig. 1A; Ji et al, 2019; Prakash et al, 2000a). Three of the seven human GBP members carry a C-terminal CaaX box motif and are post-translationally farnesylated (GBP1) or geranylgeranylated (GBP2 and GBP5), facilitating their membrane interaction (Britzen-Laurent et al, 2010; Olszewski et al, 2006). Like other dynamin superfamily members, GBPs dimerize via a highly conserved surface across the nucleotide-binding site, the "G interface," leading to stimulation of their GTPase activity (Ghosh et al, 2006). Some GBPs, e.g., the biochemically best characterized member, human GBP1, have the unique ability to hydrolyze GTP in two consecutive cleavage steps

[1]Structural Biology, Max-Delbrück-Center for Molecular Medicine in the Helmholtz Association (MDC), 13125 Berlin, Germany. [2]Institute for Chemistry and Biochemistry, Freie Universität Berlin, 14195 Berlin, Germany. [3]In Situ Structural Biology, Max-Delbrück-Center for Molecular Medicine in the Helmholtz Association (MDC), 13125 Berlin, Germany. [4]Institute of Molecular Pathogenicity, Faculty of Mathematics and Natural Sciences, Heinrich Heine University Düsseldorf, 40225 Düsseldorf, Germany. [5]Institute of Medical Microbiology and Hospital Hygiene, Medical Faculty and University Hospital Düsseldorf, Heinrich Heine University Düsseldorf, 40225 Düsseldorf, Germany. [6]Institute of Biochemistry, Faculty of Mathematics and Natural Sciences, Heinrich Heine University Düsseldorf, 40225 Düsseldorf, Germany. [7]Department of Molecular Genetics and Microbiology, Duke University, 27710 Durham, NC, USA. [8]Faculty of Chemistry and Biochemistry, Physical Chemistry I, Ruhr-University Bochum, 44801 Bochum, Germany. [9]Institute of Molecular Physical Chemistry, Faculty of Mathematics and Natural Sciences, Heinrich Heine University Düsseldorf, 40225 Düsseldorf, Germany. [10]Institute of Medical Physics and Biophysics, Charité-Universitätsmedizin Berlin, 10117 Berlin, Germany. ✉E-mail: oliver.daumke@mdc-berlin.de

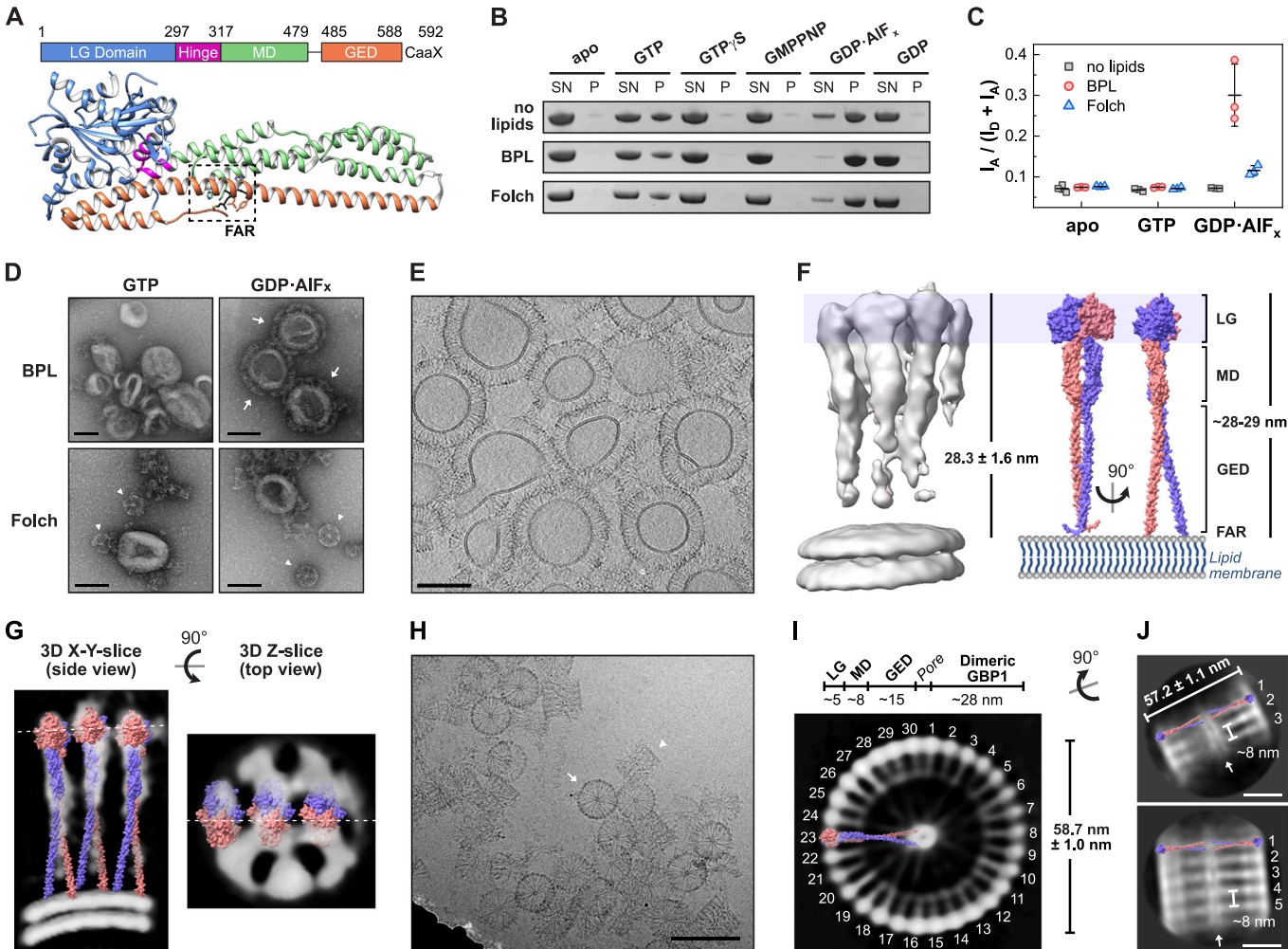

**Figure 1. Cryo-electron microscopy of oligomerized human GBP1.**

(**A**) Domain architecture of farnesylated human GBP1 (PDB 6k1z) in its nucleotide-free state. The farnesyl moiety (black, FAR) is harbored in a hydrophobic pocket (dashed box) making it inaccessible to bind to membranes (Ji et al, 2019). LG large GTPase, MD middle domain, GED GTPase effector domain. (**B**) Liposome co-sedimentation assay of GBP1 incubated with indicated nucleotides and respective liposomes of brain-derived lipid extracts. SN supernatant fraction, P pellet fraction. (**C**) Ratiometric FRET efficiencies of GBP1-Q577C-AF488 (donor) incubated with indicated liposomes supplemented with Liss Rhod PE (acceptor) and indicated nucleotides. Data from three independent replicates are shown as mean ± SD. (**D**) Negative-stain TEM of GBP1 reconstitution. Arrows indicate stable GBP1 coat, arrow heads represent soluble polymers of GBP1. Scale bars are 100 nm. (**E**) Cross section of the cyro-ET volume of GBP1-coated BPL liposomes. Scale bar is 100 nm. (**F**) Subtomogram averaging result of the GBP1 coatomer. Coatomer height (n = 110, mean ± SD) correlates with the theoretical length of an outstretched dimeric GBP1 model, generated as SWISS-MODEL (Waterhouse et al, 2018) based on the published crystal structure of GBP5-ΔGED combined with the AlphaFold2 prediction (Jumper et al, 2021; Mirdita et al, 2022) of the dimeric MD and GED of GBP1 (see Appendix Fig. S3). (**G**) Side and top view slices of the subtomogram averaging result. Dashed lines indicate the respective slice position. The outstretched dimeric GBP1 model is fitted into the protein densities. (**H**) Cryo-EM micrograph of polymeric assembly into disk-like structures. Arrow: planar disk in top view, arrow head: stacked disks in side view. Scale bar is 100 nm. (**I**) Z-slice of the GBP1 polymer 3D reconstruction. Individual building blocks are labeled and the outstretched dimeric GBP1 model is fitted into the protein densities in side view. Dimensions of individual protein domain densities are indicated and correlate with the disk diameter (n = 25, mean ± SD). (**J**) 2D classes of stacked polymeric disks. The outstretched dimeric GBP1 model is placed into the protein densities. The diameter (n = 25, mean ± SD) and the stack height are indicated. Top: three stacks, bottom: five stacks, arrow: middle pore. Scale bars are 20 nm. Source data are available online for this figure.

to guanosine monophosphate (GMP). This is achieved by repositioning the nucleotide within the nucleotide-binding pocket after the first hydrolysis step (Ghosh et al, 2006; Schwemmle and Staeheli, 1994). Whereas the first nucleotide hydrolysis step of GBP1 is essential and sufficient to restrict growth of the vacuole-residing bacterial pathogen *Chlamydia*, hydrolysis of GDP to GMP and further degradation to uric acid activates the NLRP3 inflammasome (Xavier et al, 2020).

The crystal structure of full-length GBP1 describes a closed monomeric conformation (Prakash et al, 2000a), where the farnesyl

moiety is harbored in a hydrophobic pocket (Ji et al, 2019). GTP binding promotes the formation of a head-to-head dimer via the G interface (Wehner et al, 2012). Upon GTP hydrolysis, intramolecular rearrangements induce an opening of each protomer by disrupting a salt bridge network between the LG domain and the GED, thereby releasing the farnesyl moiety (Ince et al, 2021; Sistemich et al, 2020; Vopel et al, 2010). Time-resolved Förster resonance energy transfer (FRET)-studies found that the open GBP1 conformation is stabilized by MD:MD interactions (Sistemich et al, 2020). Structural data of the GBP5 and GBP1 dimers

showed a rotation and crossover of its MDs which is stabilized via intermolecular MD:MD interactions (Cui et al, 2021; Kuhm et al, 2023). The GTP-binding and hydrolysis-induced open GBP1 conformation not only promotes membrane interaction of GBP1 dimers but also facilitates self-oligomerization into soluble tubular polymers and membrane-bound coatamers (Shydlovskyi et al, 2017; Sistemich et al, 2021, 2020). How the oligomeric interfaces between GBP1 dimers are built and how nucleotide hydrolysis coordinates the higher-order GBP1 assemblies has remained unknown.

Here, by applying cryo-electron tomography (cryo-ET) and subtomogram averaging (StA) on GBP1-coated liposomes and single particle analysis (SPA) on self-polymerized GBP1, we report an outstretched, dimeric conformation of GBP1 oligomerized on a lipid surface and in its polymeric state in solution. By characterizing the nucleotide-induced structural changes facilitating oligomerization, we address the activation mechanism of GBP1 allowing self-polymerization and coatamer formation.

## Results

### Cryo-EM reveals an outstretched dimeric conformation of oligomeric GBP1

We first aimed at the structural analysis of the GBP1 coatamer on the surface of liposomes. Protein coat formation was analyzed on liposomes derived from two brain lipid mixtures (Folch extract and Brain Polar Lipid extract, BPL) using liposome co-sedimentation assays. Since the GED opening of GBP1 including the release of its farnesyl moiety strongly depends on binding and hydrolysis of GTP (Shydlovskyi et al, 2017; Sistemich et al, 2021, 2020), we tested for liposome binding in the presence of different nucleotides (Fig. 1B). We observed sedimentation of GBP1 in the presence of GTP and the GTP transition state analog, GDP•AlF$_x$, but not in the presence of GDP or the non-hydrolysable GTP analogs, guanosine 5'-O-[γ-thio]triphosphate (GTPγS) and guanosine-5'-[(β,γ)-imido]triphosphate (GMPPNP). Sedimentation was not dependent on the presence of liposomes, as reported previously (Fres et al, 2010; Shydlovskyi et al, 2017), suggesting the formation of non-membrane-bound, "soluble" polymers which can also be sedimented (Fig. 1B).

To distinguish between liposome binding and liposome-independent polymer formation, we employed a FRET-based liposome-binding assay with donor-labeled GBP1 and acceptor-labeled liposomes. Prominent changes in the FRET efficiency indicating membrane binding were only observed when BPL liposomes and GDP•AlF$_x$ were used (Fig. 1C). At GBP1 concentration greater than 10 µM, the FRET signal saturated indicating full coating of the liposomes (Appendix Fig. S1A). Indeed, in negative-stain transmission electron microscopy (TEM), a stable protein coat was only observed for BPL liposomes in the presence of GDP•AlF$_x$ (Fig. 1D). In rare cases, we additionally observed soluble polymers attached to protein-coated BPL liposomes (Appendix Fig. S2). In contrast, in the presence of liposomes constituted by Folch lipids, liposome-independent polymer formation was favored over membrane binding. While GBP1 binding to liposomes could not be rescued by supplementing the Folch extract with negatively charged lipids

(10% phosphatidylinositol-4,5-bisphosphate or 10% phosphatidylserine), addition of 25% cholesterol efficiently restored membrane binding (Appendix Fig. S1B). This may indicate that high membrane fluidity and elasticity promote GBP1 binding to liposomes.

The GDP•AlF$_x$-bound GBP1 coatamer was well recognizable in cryo-ET volumes of BPL liposomes (Fig. 1E). The protein coat had an average height of 28.3 nm ± 1.6 nm, which agrees well with the theoretical dimension of a dimeric, outstretched GBP1 dimer (~28.8 nm, Appendix Fig. S3). We further performed subtomogram averaging which resulted in a map of membrane-bound GBP1 dimers at a moderate resolution of ~26 Å (Figs. 1F and EV1, Movie EV1, Appendix Fig. S4). The dimension of the density corresponded to the outstretched conformation of dimeric GBP1. Although the map did not allow for a detailed molecular fitting of the GBP1 dimer, it suggested the potential oligomerization interfaces: the MD and GED were apparently not involved in higher order assembly, based on the absence of density between these elements outside the dimer (Fig. 1G). However, we cannot exclude that potentially existing, weak interactions between these domains may have been averaged out as a result of a high flexibility within the oligomer. In contrast, the dimeric LG domains displayed lateral contacts mediating the formation of a two-dimensional protein lattice.

In the absence of lipid membranes and presence of GDP•AlF$_x$, GBP1 stably polymerized into disk-like structures with an outer diameter of 58.7 nm ± 1.0 nm (Fig. 1H, Appendix Fig. S5A), consistent with recently reported findings (Kuhm et al, 2023; Shydlovskyi et al, 2017). Aiming to obtain an overview of the polymeric assembly, we performed single particle cryo-EM imaging. Particles of polymeric disks were highly heterogeneous. We obtained a low-resolution 3D reconstruction at a resolution of ~37 Å, which provided a basic understanding of the disks' architecture (Figs. 1I and EV2). 30 subunits arrange into a planar ring-like structure and show a prominent central density connecting to the outer ring via faint protein densities. The distance between the outer and inner ring was ~28 nm, resembling the dimension of the dimeric GBP1 model in its open outstretched conformation. In this orientation, the farnesyl moieties constitute the central ring-like density with a pore size of few nanometer diameter, whereas the LG dimers assemble in the peripheral ring. Faint protein densities next to one LG dimer match the dimension of the MD (~8 nm) and diffuse densities between the MD and the prominent density of the C-terminal farnesyl ring match the dimensions of the GED (~15 nm), suggesting that GBP1 dimers in the outstretched conformation are the building blocks of the polymers. The model further indicates that interdimeric interfaces are established between LG domains laterally. 2D classification of short tubular structures in side views further shows that larger oligomeric assemblies are apparently formed by stacking of planar disks (Fig. 1J). The distance between two disks was ~8 nm which agrees well with the dimension of two stacked GBP1 dimers (Appendix Fig. S5B). Although the largest 3D class from classification supports a stacking mechanism, minor populations within 2D classes appear offset, suggesting that at least some particles or regions have helical symmetry.

Together, both the membrane-bound and polymeric state reveal an outstretched dimeric conformation of GBP1, and regions at the periphery of the LG domain dimer contribute to the higher-order oligomeric assembly.

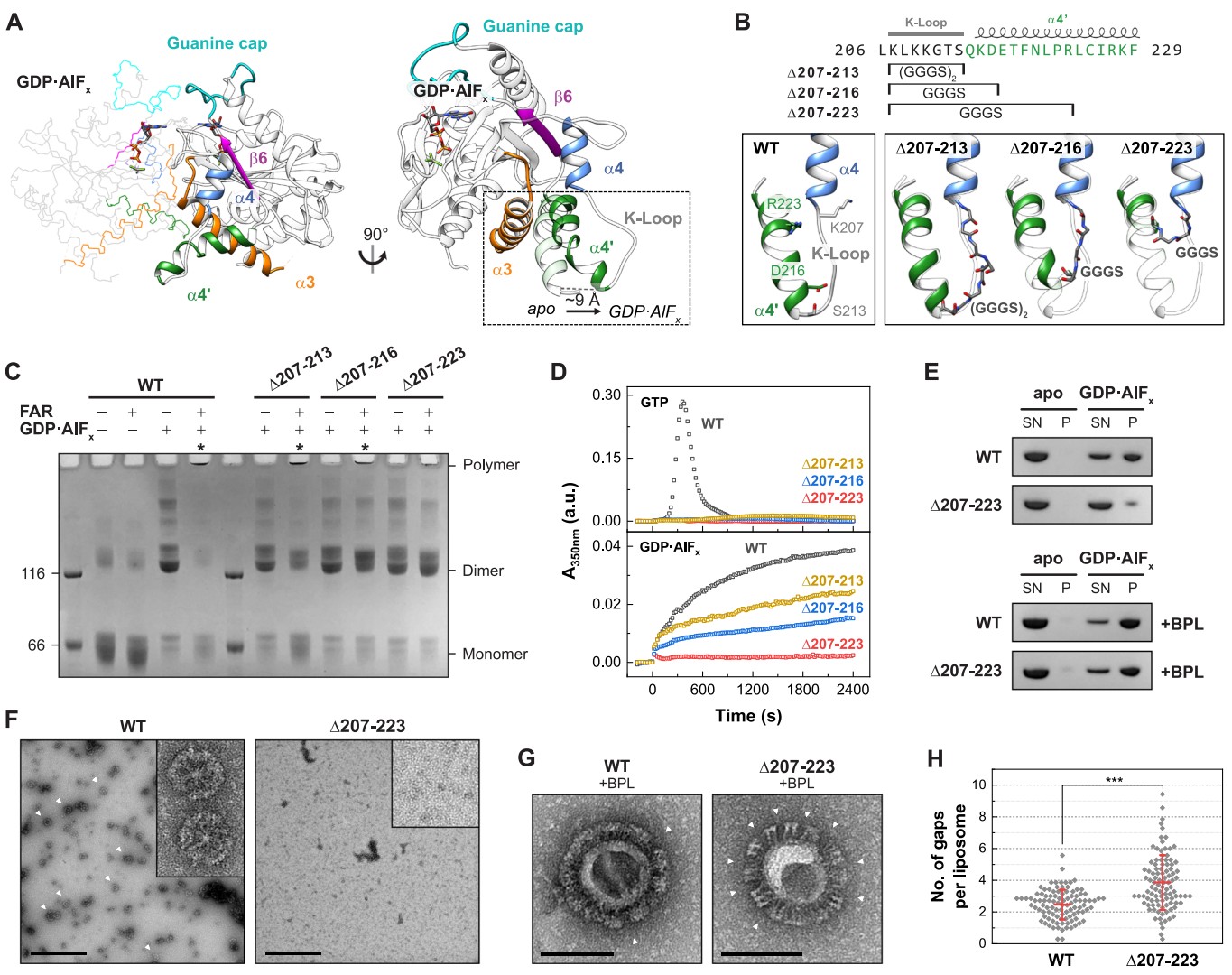

**Figure 2. Oligomerization-deficient helix α4' variants.**

(A) Structure of the LG dimer (left) and superimposition of the GDP•AlF$_x$-bound (PDB 2b92) and the apo state (PDB 1dg3, transparent). (B) Design of helix α4' variants. Sequence of GBP1 with respective constructs marked (top). Close view of helices α4, α4', and the K-Loop with start (K207) and respective end positions (S213, D216, R223) of wild-type GBP1 (lower left), and AlphaFold2 predictions (Jumper et al, 2021; Mirdita et al, 2022) of the constructs showing the respective regions replaced by GGGS-linkers (lower right). Δ207-213 contains two GGGS repeats due to a longer distance. (C) Crosslinking assay of helix α4' variants. The respective oligomeric states based on molecular weight are indicated. Asterisks indicate polymerization. FAR farnesylated. (D) Light scattering-based polymerization assay of helix α4' variants. Polymerization is induced by GTP (top) and GDP•AlF$_x$ (bottom). (E) Sedimentation assay of oligomerization-deficient Δ207-223 mutant in the absence (top) and in presence (bottom) of BPL liposomes. SN supernatant fraction, P pellet fraction. (F) Negative-stain TEM of polymeric disk formation (arrow heads). Scale bars are 500 nm. (G) Negative-stain TEM of GBP1-coated liposomes. Arrow heads indicate gaps within the patched protein coat. Scale bars are 100 nm. (H) Quantification of gaps within the protein coat per liposome ($n = 100$) in negative-stain TEM micrographs in a blinded experiment. Data are averages from seven independent experimenters and are represented by mean ± SD. Significance was derived by an unpaired $t$ test. ***$P \leq 0.001$. See also Appendix Fig. S8. Source data are available online for this figure.

## The peripheral helix α4' in the LG domain mediates oligomerization

To delineate the oligomerization interfaces within the GBP1 oligomer, we re-analyzed the previously described structural transitions from the monomeric apo to the dimeric, assembly-competent GDP•AlF$_x$-bound state (Ghosh et al, 2006; Prakash et al, 2000b). A prominent structural rearrangement was observed for helix α4' at the distal side of the G-interface that shifts by ~9 Å to avoid a steric clash with the opposing LG domain (Fig. 2A).

We probed whether the observed motion might also generate an oligomeric contact site upon dimerization. Hence, based on structural predictions, we designed GBP1 variants in which specific parts of the protruding helix α4' were exchanged with a glycine-serine linker (Fig. 2B). Analytical size-exclusion chromatography and circular dichroism (CD) measurements confirmed that the overall protein structure was maintained in the helix α4' variants (Appendix Fig. S6). We analyzed the ability of these three variants to dimerize and polymerize using a crosslinking-based approach (Fig. 2C). Without nucleotide (apo), both non-farnesylated and

farnesylated wild-type protein remained in a monomeric state. Addition of GDP•AlF$_x$ led to dimerization of all non-farnesylated constructs, whereas farnesylated wild-type protein showed a dramatic shift toward higher molecular weight species at the top of the gel. In addition, the monomeric and dimeric states vanished for wild-type GBP1, suggesting that this species represents the crosslinked disks (Fig. 1H–J). Like wild-type, a variant with a Gly-Ser substitution of a lysine-rich loop (the K-loop) preceding helix α4' polymerized in the presence of GDP•AlF$_x$ (construct Δ207-213), and some dimeric species were detected. When further substituting parts of helix α4', the equilibrium shifted from polymers to dimers, suggesting that GDP•AlF$_x$-induced polymerization, but not dimerization, was impaired. While the GBP1 Δ207-216 variant showed reduced polymer formation, polymerization of the Δ207-223 variant was mostly abolished.

Using a light scattering-based approach, we further monitored the polymerization of farnesylated GBP1 in the presence of GTP and GDP•AlF$_x$ over time (Fig. 2D). In the presence of GTP, wild-type GBP1 showed typical polymerization kinetics with an initial lag phase, which was followed by a strong increase and decrease in light scattering, due to the GTP hydrolysis-dependent assembly and disassembly of oligomeric structures (Shydlovskyi et al, 2017; Sistemich et al, 2020; Fig. 2D). In contrast, the three mutants showed only small increases in light scattering under these conditions. In the presence of GDP•AlF$_x$, farnesylated wild-type GBP1 steadily polymerized over a time span of 40 min, representing the slow formation of disk-like structures (Shydlovskyi et al, 2017; Fig. 1H–J). In accordance with the crosslinking assay, both Δ207-213 and Δ207-216 variants polymerized in the presence of GDP•AlF$_x$, but with slower kinetics. In contrast, GBP1 Δ207-223 did not polymerize, suggesting a crucial role of helix α4' in oligomer formation.

These findings were further supported by sedimentation experiments and negative-stain TEM analysis. While wild-type GBP1 was sedimented in the presence of GDP•AlF$_x$, the vast majority of mutant Δ207-223 remained in the supernatant (Fig. 2E). Using negative-stain TEM, we observed polymeric disks for wild-type GBP1, but not for the GBP1 Δ207-223 mutant (Fig. 2F).

We next investigated the effect of the Δ207-223 substitution on the formation of the GBP1 coatomer. In the monomeric state, helix α4' is involved in an intramolecular LG:GED interaction via a salt bridge network between R227/K228 and the four glutamate residues 556, 563, 568, and 575 within helix α12/13 (Vopel et al, 2010; Appendix Fig. S7). We therefore asked whether the GBP1 Δ207-223 variant can adopt the open dimeric state with an accessible farnesyl moiety. Liposome co-sedimentation experiments revealed that both wild-type GBP1 and the Δ207-223 variant bound to BPL liposomes (Fig. 2E), implying that our mutant like wild-type GBP1 has an accessible farnesyl anchor and can exist in an open dimeric conformation. Interestingly, the GBP1 Δ207-223 variant failed to establish a uniform protein coat on BPL liposomes, but rather formed protein patches on liposomes resulting in a high number of gaps within the protein coat (Fig. 2G,H, Appendix Fig. S8). This observation suggests that weaker interactions between GBP1 dimers prevent the formation of a stable protein coat. Hence, helix α4' is not only crucial for the oligomerization of soluble GBP1 polymers, but also for the formation of a uniform and stable protein coat on membranes.

## Helix α4' is crucial for GDP hydrolysis and GBP1 binding to pathogenic bacteria

Since GBP1 polymerization is dependent on GTP and GDP hydrolysis (Fig. 2D; Shydlovskyi et al, 2017), we determined GTP binding affinities and GTPase activities of the helix α4' mutants. Affinities of the helix α4' mutants for binding a non-hydrolysable, fluorescently labeled GTP analog resembled that of wild-type GBP1 (Figure EV3A). Furthermore, all helix α4' mutants retained the ability to hydrolyze GTP; however, GDP hydrolysis and thus GMP production was abolished for all of them (Fig. 3A,C). Wild-type GBP1 showed a concentration-dependent increase in specific GTPase activity that can be explained with a dimerization-dependent, cooperative hydrolysis mechanism (Ghosh et al, 2006). The apparent $K_d$ for wild-type GBP1 dimer formation was 1.9 μM with a maximum catalytic GTPase activity, $k_{max}$, of 55.3 min$^{-1}$. The catalytic GTPase activity of the Δ207-223 variant was increased 1.7-fold compared to wild-type, presumably due to the lack of steric restrains within the dimer, while the apparent $K_d$ of the mutant resembled that of wild-type (Fig. 3B). Also, the GBP1 Δ207-213 and Δ207-216 variants retained a similar cooperative GTPase activity to wild-type GBP1 with a slightly reduced dimerization affinity (1.8-fold and 3.0-fold lower affinity for Δ207-216 and Δ207-213, respectively).

While full-length GBP1 cannot utilize GDP when provided as a substrate, but only from preceding GTP hydrolysis, the isolated LG domain lacking the auto-inhibiting GED shows GDP-induced dimerization and an increased GDPase activity (Ince et al, 2021). To explore why the helix α4' mutants produce less GMP than wild-type GBP1, we determined assembly and GDPase activity of the isolated LG domains. As expected, the wild-type LG domain eluted as a monomer in the absence of nucleotide (apo), but efficiently dimerized in the presence of GDP•AlF$_x$ and GMP•AlF$_x$, mimicking the first and second GTP hydrolysis step, respectively. Also, the α4' variants were monomeric in the apo state and dimerized in the presence of GDP•AlF$_x$. However, they did not form dimers in the presence of GMP•AlF$_x$ (Fig. 3D). When offering GDP as substrate, the α4' variant LG domain constructs showed a drastically reduced catalytic GDPase activity as compared to the wild-type LG domain (Fig. 3E), even though the binding affinities toward mant-GDP resembled that of wild-type (Figure EV3B). These data indicate that GTP-induced dimerization and the subsequent GTP hydrolysis is not affected by mutations in helix α4'. However, mutations in α4' promote LG dimer dissociation upon the first GTP hydrolysis step.

We next aimed to characterize the relevance of helix α4' on the binding to a gram-negative bacterial pathogen. While wild-type GBP1 formed a stable protein coat around pathogenic E. coli, the mutants showed reduced capacity in encapsulating bacteria (Fig. 3F,G, Movie EV2). Even though the Δ207-223 variant was still able to bind to artificial liposomes, it completely failed to form a protein coat on pathogenic E. coli.

## A coordinated movement of helix α4' and α3 mediates nucleotide hydrolysis

Further analyzing the nucleotide-dependent structural transitions of the GBP1 LG domain for the available nucleotide states (Ghosh et al, 2006; Prakash et al, 2000b), we observed a coupled motion of

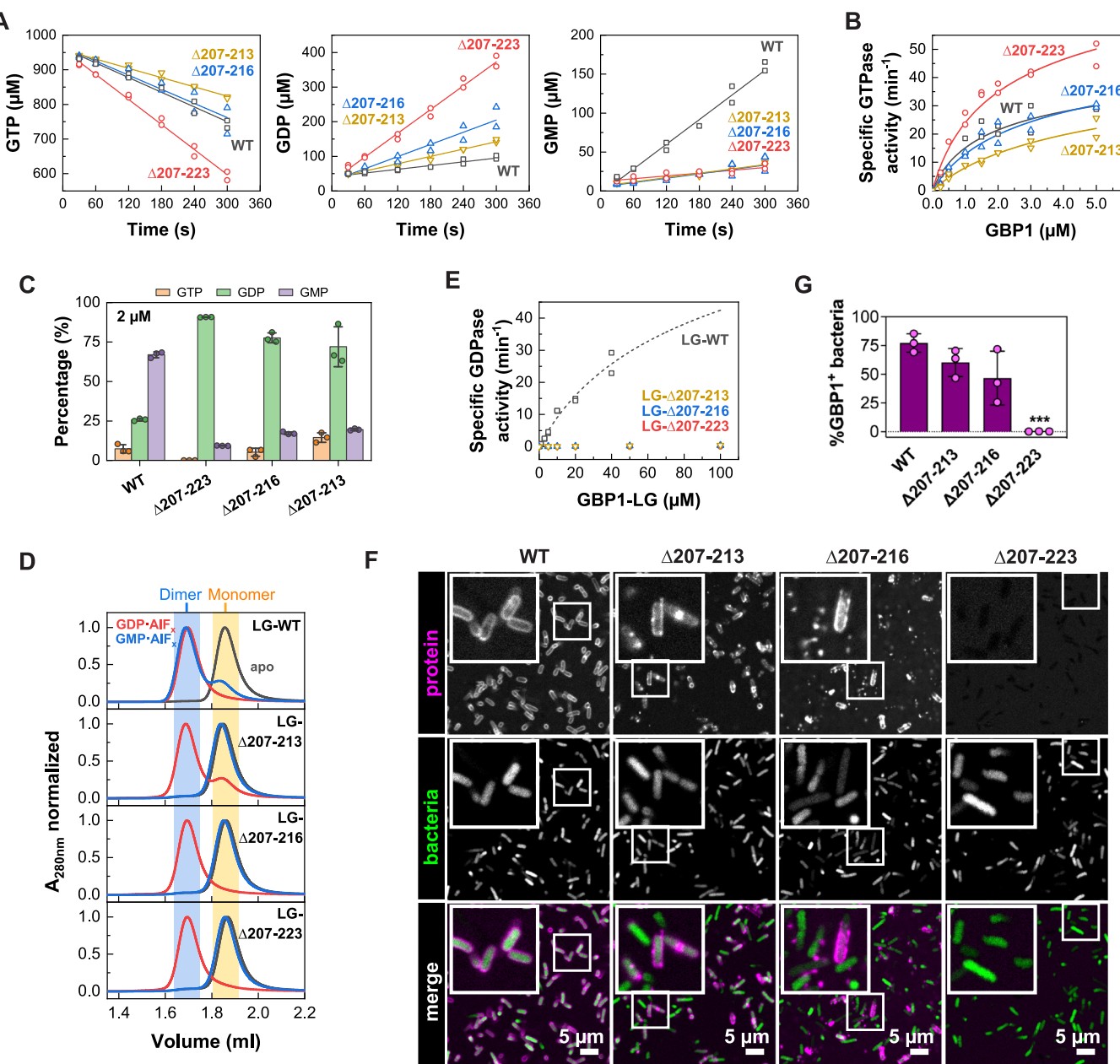

**Figure 3. Helix α4′ is crucial for GDP hydrolysis and GBP1 binding to pathogenic bacteria.**

(A) GTP hydrolysis (left), GDP production (middle), and GMP production (right) of 2 μM wild-type GBP1 and respective helix α4′ variants derived from two independent experiments. (B) Specific activity of cooperative GTP hydrolysis. Initial hydrolysis rates ($n = 2$) were normalized to the protein concentration and plotted against protein concentration. Dimer dissociation constants, $K_d$, and maximum catalytic GTPase activities, $k_{max}$, were calculated by fitting a quadratic equation to data using Eq. 2. (C) End product formation of GTP hydrolysis for GBP1 helix α4′ constructs after 30 min ($n = 3$, mean ± SD). (D) Analytical size-exclusion chromatography of isolated LG domains. Dimeric fractions are highlighted in blue, monomeric fractions in yellow. (E) Specific activity of cooperative GDP hydrolysis of isolated LG domains. Initial hydrolysis rates ($n = 2$) were normalized to the protein concentration and plotted against protein concentration. The dashed line represents a fit for WT using Eq. 2 ($K_d = 110$ μM ± 50 μM, $k_{max} = 120$ min⁻¹ ± 40 min⁻¹). (F) Confocal microscopy images of wild-type GBP1 and helix α4′ variants targeting pathogenic *E. coli*. Representative WT images in Figs. 3F and EV4C,F are identical and derived from the same experiments performed with all constructs in parallel. (G) Quantification of GBP1-targeted (GBP1⁺) bacteria in in vitro binding assay after 30 min. Graphs are averages from three independent experiments and are represented by mean ± SD. One-way ANOVA with Dunnett's multiple comparisons test comparing to GBP1 WT⁺ bacteria was used, all statistically significant comparisons are shown. ***$P < 0.001$. WT data in Figs. 3G, 4G, and EV4D are derived from the same experiments performed with all constructs in parallel and provided here for comparison. Source data are available online for this figure.

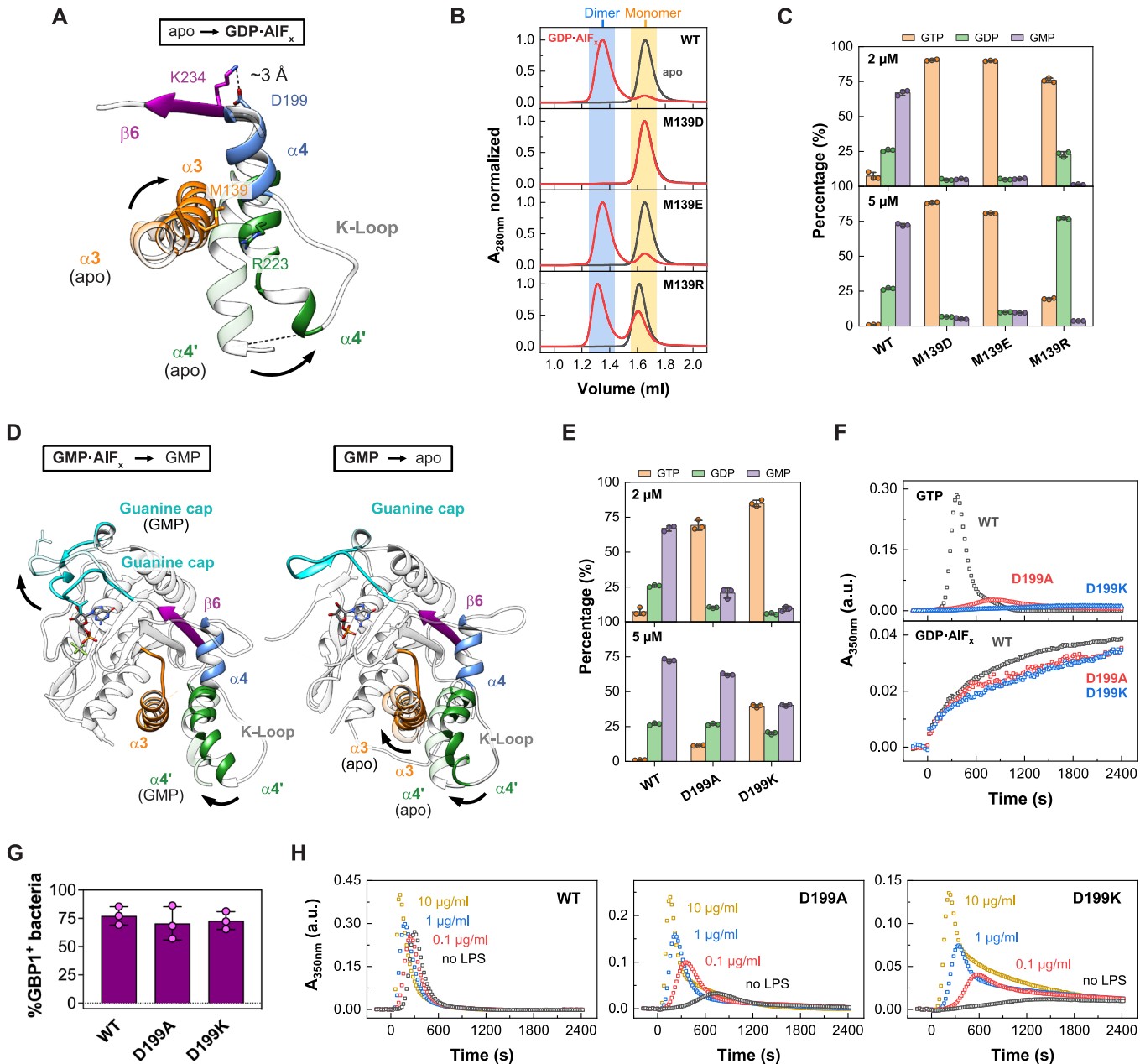

**Figure 4. Intramolecular movements mediate nucleotide hydrolysis and control oligomerization.**

(A) Superimposition of GBP1 LG domain from the apo (PDB 1dg3, transparent) to the GDP•AlF$_x$-bound state (PDB 2b92). (B) Analytical size-exclusion chromatography of M139 mutants. Dimeric fractions are highlighted in blue, monomeric fractions in yellow. (C) End product formation of GTP hydrolysis for M139 mutants after 30 min (*n* = 3, mean ± SD). (D) Left: Transition of the GBP1 LG domain from the GMP•AlF$_x$-bound (PDB 2b8w) to the GMP-bound state (PDB 2d4h, transparent). Right: Transition of the GBP1 LG domain from the GMP-bound (PDB 2d4h) to the apo state (PDB 1dg3, transparent). (E) End product formation of GTP hydrolysis for GBP1 pivot point mutants after 30 min (*n* = 3, mean ± SD). (F) Light scattering-based polymerization assay of pivot point mutants. Polymerization is induced by GTP (top) and GDP•AlF$_x$ (bottom). WT data are the same as shown in Fig. 2D and provided here for comparison. (G) Quantification of GBP1-targeted pathogenic *E. coli*, as described in the legend of Fig. 3G. See Figure EV4B for the confocal images. No significant statistical changes were observed. (H) Light scattering-based polymerization assay of pivot point mutants in presence of LPS. Polymerization is induced by GTP. Source data are available online for this figure.

helix α4' and the adjacent helix α3, the latter of which contributes to the dimerization interface (Fig. 4A, Movies EV3 and EV4). This suggested that dimerization-dependent structural changes are coordinated between helices α3 and α4'.

To evaluate this hypothesis, we designed two mutants to lock helix α4' to helix α3 either in a closed state, as observed in the monomeric apo and GMP-bound states, or in a more open conformation, as in the dimeric GMP•AlF$_x$-bound, GDP•AlF$_x$-bound, and GMPPNP-bound conformations. For the locked state, we mutated Met139 in helix α3 into an aspartate which should form a new salt bridge to Arg223 within helix α4' (Fig. 4A, Movie EV4). To stabilize helix α4' in an open state, we mutated Met139 to

a glutamate which, due to its longer side chain compared to the aspartate, should introduce a greater distance between the helices. Additionally, by mutating Met139 into an arginine, we designed a mutant to unlock helix α3 and helix α4' due to repulsion with the opposing Arg223.

Analytical size-exclusion chromatography of the non-farnesylated mutants validated GDP•AlF$_x$-dependent locking of M139D in the monomeric state, while M139E was indeed able to dimerize (Fig. 4B). Also, the M139R mutant dimerized, although with reduced efficiency. While all three mutants showed similar nucleotide affinities compared to the wild-type protein (Figure EV3A), GTP hydrolysis of M139D and M139E was completely abolished (Fig. 4C); even though preventing the release of the GED by crosslinking to helix α4' still allowed for GTP but not GDP hydrolysis (Ince et al, 2021). The M139R mutant, however, hydrolyzed GTP but failed to hydrolyze GDP, suggesting a specific deficit in the second hydrolysis step.

Interestingly, in the farnesylated state, GDP•AlF$_x$-dependent dimerization and higher order oligomerization of the dimerization-capable mutants M139E and M139R were blocked (Figure EV4A,B). Accordingly, these mutations also abolished binding to pathogenic *E. coli* and prevented the formation of a stable protein coat (Figure EV4C,D). We concluded that when the coordinated α3-α4' motion is hindered as in our mutants, farnesylated GBP1 fails to adopt an open conformation, and thus, oligomerization and coatomer formation are prevented. A correctly coordinated α3-α4' motion is therefore required for releasing the farnesyl-stabilized GED from helix α4' (Appendix Fig. S7), allowing GBP1 to fulfill its biological function in innate immunity.

While a coordinated α3-α4' movement was apparent upon GTP-induced dimerization (Movies EV3, Figure EV4), helix α4' moves independently of helix α3 from the dimeric GMP•AlF$_x$- to the GMP-bound monomeric state (Fig. 4D, Movie EV4). Instead, a simultaneous motion of helix α4' and the guanine cap was observed between the two states, hinting at a long-range conformational coupling of these two elements during GDP hydrolysis (Fig. 4D, Movie EV3). We identified a salt bridge between Lys234 of the β6 strand and Asp199 located in helix α4, which may couple the motions of helix α4' with the guanine cap (Fig. 4A,D, Movies EV3 and EV4). This was reminiscent of a lever motion, with the salt bridge of Lys234-Asp199 acting as the pivot point.

To explore a possible function of the pivot point, we disrupted the salt bridge by mutating the Asp199 to either an alanine or lysine. Although there was no significant difference in nucleotide affinity (Figure EV3A), both mutations led to a dramatic reduction in GTP hydrolysis and GMP production as compared to wild-type (Fig. 4E). The mutants showed strongly reduced polymerization upon GTP addition, while dimerization and polymerization for GDP•AlF$_x$-induced assembly were not restricted (Figs. 4F and EV4E), indicating that the oligomeric interface via helix α4' can still be established. In support of this hypothesis, we found that the pivot point mutants encapsulated pathogenic *E. coli* with similar efficiencies as wild-type GBP1, suggesting that LPS present on the bacterial surface stabilizes GBP1 oligomerization (Kutsch et al, 2020; Figs. 4G and EV4F). In line with this observation, addition of LPS facilitated GTP-induced polymerization of the pivot point mutants in a concentration-dependent manner (Fig. 4H). Together, our studies and the published structural data suggest that the nucleotide-loading status of the LG domain coordinates movements of the guanine cap and/or helix α3-α4' via a pivot point, allowing GBP1 oligomerization.

## Discussion

Here, we have synthesized new and known structural data of GBP1 with biochemical and mechanistic experiments to derive GBP1's biological function in innate immunity. We delineate the activation mechanism for GBP1 oligomerization. By assembling into large polymers and a membrane-bound protein coat, GBP1 plays a crucial role in the innate immune response against microbial pathogens. We show that the peripheral helix α4' in the LG domain is critical for establishing the oligomeric interface and facilitates the formation of an antimicrobial protein coat. Our studies reveal that coordinated movements of structural elements in the LG domain are a prerequisite for a nucleotide-driven activation mechanism allowing oligomerization and membrane binding.

Our cryo-ET and cryo-EM data demonstrate that human GBP1 binds to lipid membranes in an open, outstretched conformation. The observed preference of membrane-independent oligomerization for Folch liposomes versus BPL may reflect the consumption of soluble oligomers at the expense of coatomers in the presence of BPL liposomes. Within the coatomer, the overall shapes of the LG domain and MD match well with the recently published crystal structure of the GBP5 dimer (Cui et al, 2021) and the cryo-EM structure of the GBP1 dimer (Kuhm et al, 2023), suggesting that the coatomer is comprised of dimeric building blocks with crisscrossed MDs that are extended by elongated GEDs. The GEDs extend in parallel toward the membrane surface, allowing the C-terminal farnesyl moiety to insert into the lipid bilayer. The observed conformation is in accordance with the theoretical dimension of outstretched GBP1 and with previous biophysical and structural studies predicting an upright orientation of GBP1 on membranes (Kuhm et al, 2023; Sistemich et al, 2021) and on *Salmonella* minicells (Zhu et al, 2021), and within soluble polymers (Kuhm et al, 2023; Peulen et al, 2023; Shydlovskyi et al, 2017; Sistemich et al, 2020).

Our structure-function approach demonstrates that the peripheral helix α4' of the LG domain is crucial for the assembly of GBP1 dimers into polymeric structures and forming a uniform protein coat on lipid membranes. In the absence of lipid membranes, outstretched GBP1 dimers polymerize into planar disk-like structures of 30 building blocks. Polymeric disks, in turn, might stack into larger tubular structures, as also reported recently (Kuhm et al, 2023; Shydlovskyi et al, 2017; Sistemich et al, 2020). Our experiments suggest that lateral LG domain interactions observed for GBP1 in the membrane-bound coatomer may also stabilize interactions within soluble polymers.

Our data and the available structural and functional information provide the molecular basis for a nucleotide-driven activation mechanism in GBP1 governing the motions of helix α4', helix α3, the guanine cap, and the GED, thereby coordinating the nucleotide-loading state with oligomerization and membrane binding (Fig. 5): Nucleotide-free GBP1 exist in an auto-inhibited, monomeric form where the GED is locked to helix α4' in a closed conformation and the farnesyl moiety is inaccessible (Ji et al, 2019; Prakash et al, 2000a). The guanine cap is in an open conformation, whereas helix α4' is in a locked conformation, thereby blocking the G-interface in the LG domain (Prakash et al, 2000a). GTP-binding to GBP1 induces a simultaneous closing of the guanine cap and a

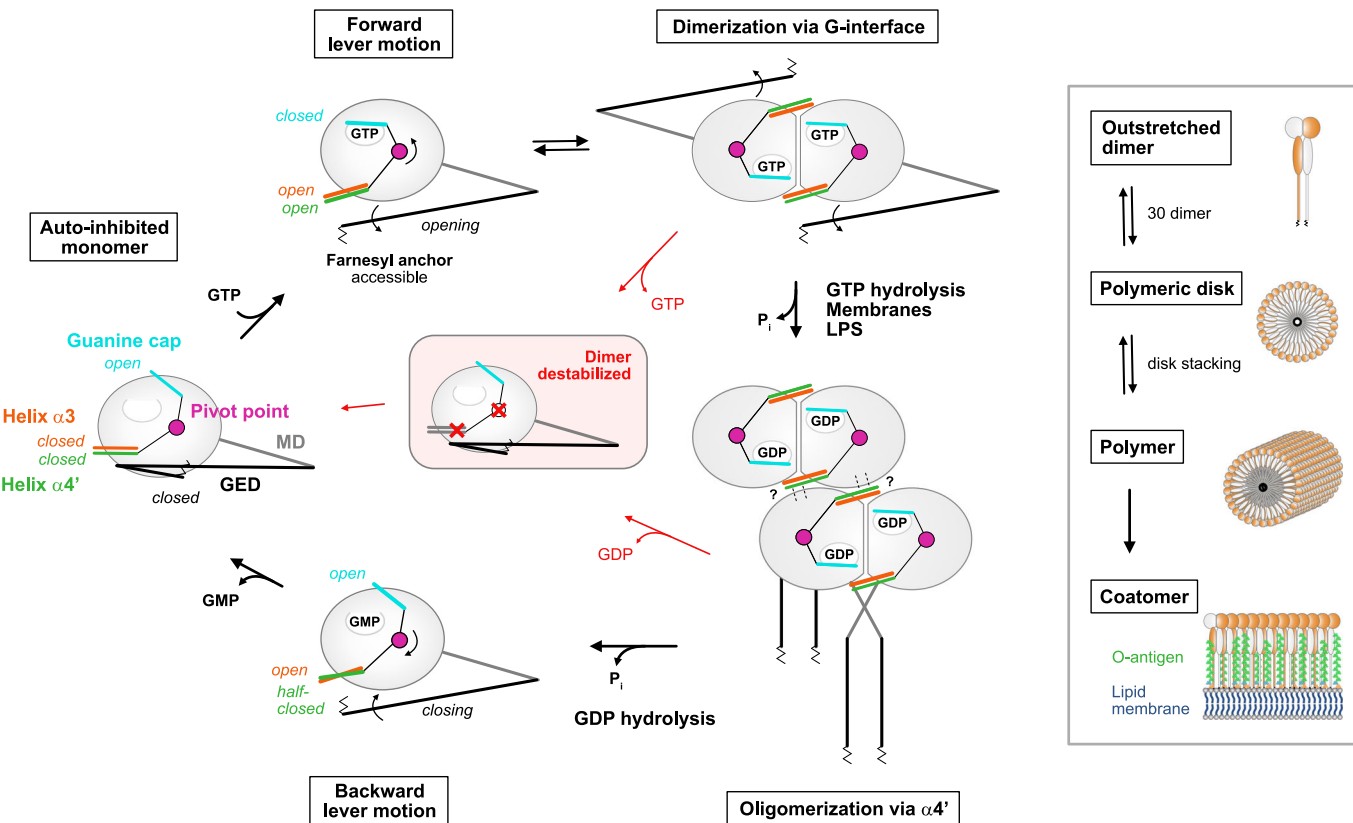

**Figure 5. Model of the nucleotide-driven activation mechanism in GBP1.**

GTP-binding induces a simultaneous closing of the guanine cap and a coordinated opening of helix α3-α4' via the salt bridge D199-K234 (pivot point), allowing LG dimerization and GTPase activation. Helix α4' motion releases the GED from the LG domain and the farnesyl moiety becomes accessible. GBP1 dimerizes in a crisscrossed conformation and lateral LG interactions via helix α4' build the oligomeric interface. Dimeric GBP1 in an outstretched conformation is the building block of higher-ordered assemblies. 30 dimers assemble into planar polymeric disks, that stack into large soluble polymers. Polymers are required to establish a protein coat (coatomer) on bacterial pathogens. Following GDP hydrolysis, GBP1 dimers dissociate, the guanine cap opens, and helix α4' partially closes. Upon nucleotide dissociation, helix α3 together with helix α4' take a closed conformation. Interference with the overall lever motion or impairing the lever arms destabilizes the GBP1 dimer leading to dimer dissociation. LG large GTPase, MD middle domain, GED GTPase effector domain.

coordinated opening of helix α3-α4' which we interpret as forward lever motion via the identified pivot point in the LG domain. These conformational changes allow for LG dimerization and GTPase activation (Ghosh et al, 2006; Prakash et al, 2000b; Wehner et al, 2012). At the same time, helix α4' pushes the GED away from the LG domain thereby releasing the farnesyl moiety from its pocket and facilitating the crisscross arrangements of the stalks (Cui et al, 2021; Ghosh et al, 2006; Ince et al, 2021; Kuhm et al, 2023; Sistemich et al, 2020; Vopel et al, 2010). The farnesyl moiety can insert into the membrane (Britzen-Laurent et al, 2010), while helix α4' is available to form a stable oligomeric interface via the LG domains. GTPase-induced movements of helix α4' between the GMPPNP- and GDP•AlFₓ-bound X-ray structures are rather minor, yet may further promote the oligomeric assembly via helix α4' (Ghosh et al, 2006). Following GDP hydrolysis, the GBP1 dimer dissociates in the GMP-bound state concomitant with an opening of the guanine cap and a partial closing of helix α4', while helix α3 remains in an open conformation. Upon nucleotide dissociation, helix α3 and α4' synchronously move back toward a closed conformation which we consider as backward motion of the lever arm. Thus, helix α4' has multiple functions in the GTPase cycle of GBP1: Preventing LG dimerization and GED opening in the

nucleotide-free auto-inhibited state, while it promotes GED opening and allows LG dimerization and oligomerization in the activated state.

The identified pivot point is conserved within the GBP, but not the closely related atlastin protein family (Figure EV5). Interference with the pivot point appears to decouple the overall lever motion between the guanine cap and the combined motion of helix α4' and helix α3. Both helices are in close proximity with the dimer interface and likely destabilize the GTP-bound GBP1 dimer, abrogating GTPase activation and thus affecting oligomerization. Impairing the lever arm, e.g., by altering helix α4' in the Δ207-223, Δ207-216, and Δ207-213 mutants, by unlocking helices α3-α4' in the M139R mutant, or by covalently locking the GED to helix α4' (Ince et al, 2021), abolishes GMP production. As these constructs interfere with the interplay between helix α4' and α3, we hypothesize that the lever mechanism via the pivot point still allows for GTPase activation, but the GDP-bound dimer is destabilized. Accordingly, dimer dissociation is favored over consecutive GDP hydrolysis.

As reported, polymeric GBP1 directly binds to lipopolysaccharide (LPS) and transitions into an LPS-stabilized protein coat on the bacterial surface (Kutsch et al, 2020). Severely weakening the lateral interactions via helix α4', thus abolishing its polymerization as in the Δ207-223 variant, completely prevented encapsulation of gram-

negative bacteria. When restricting the assembly of higher-ordered polymers but not the formation of polymeric disks, as in the Δ207-213, Δ207-216 variants and the pivot point mutants, GBP1 still established a coatomer on pathogens. This might be explained by an LPS-stabilized assembly mechanism of higher-ordered polymers on the surface of the pathogens (Dickinson et al, 2023; Kutsch et al, 2020), which also compensates for the reduced oligomerization efficiency of the pivot point mutants.

In summary, our structure-function study elucidates the activation mechanism of GBP1, therefore deepening our understanding of the underlying molecular coupling of the GTPase cycle and oligomerization within the GBP protein family which is crucial for its antimicrobial functions.

# Methods

## Reagents and tools

See Table 1.

## Methods and protocols

### Protein expression and purification

Expression, purification, and farnesylation were performed as recently described (Sistemich and Herrmann, 2020). Briefly, wild-type GBP1 and all constructs (generated by site-directed mutagenesis) were expressed from pQE-80L vector as N-terminal His$_6$-tag fusions in BL21-CodonPlus (DE3)-RIL or BL21 (DE3) cells, respectively. For structural studies, non-farnesylated wild-type GBP1 was purified by affinity chromatography (HisPur Cobalt Resin, Thermo Fisher Scientific) followed by size-exclusion chromatography (Superdex 200 26/600, GE Healthcare). Farnesylation by enzymatic modification was performed using 60 μM GBP1, 1.25 μM self-prepared FTase, and 150 μM farnesyl pyrophosphate (FPP) in 50 mM Tris-HCl pH 7.9, 150 mM NaCl, 5 mM MgCl$_2$, 10 μM ZnCl$_2$ in a glass vial at 4 °C overnight. FTase was purified according to Dickinson et al (2023). Farnesylated and non-farnesylated GBP1 were separated by hydrophobic interaction chromatography (Butyl Sepharose HP, GE Healthcare), followed by a second size-exclusion chromatography run in

**Table 1. Reagents and tools.**

| Reagent or resource | Source | Identifier |
|---|---|---|
| **Bacterial strains** | | |
| BL21 (DE3) | Novogene | N/A |
| BL21-CodonPlus (DE3)-RIL | Stratagene | N/A |
| DH5-Alpha | Novogene | N/A |
| *Escherichia coli* DSM 1103 | DSMZ | N/A |
| **Chemicals** | | |
| Bis(sulfosuccinimidyl)suberate (BS$^3$) | Thermo Fisher Scientific | A39266 |
| Farnesyl pyrophosphate (FPP) | Cayman Chemical | 63250 |
| Alexa-Fluor 488 C5 Maleimide (AF488) | Thermo Fisher Scientific | A10254 |
| Alexa-Fluor 647 C2 Maleimide (AF647) | Thermo Fisher Scientific | A20347 |
| Lipopolysaccharides from *E. coli* O55:B5 (LPS-O55:B5) | Invivogen | tlrl-pb5lps |
| Brain Polar Lipid (BPL) Extract (porcine) | Avanti Polar Lipids | 141101 |
| Brain Extract from bovine brain, Type I, Folch Fraction I | Sigma-Aldrich | B1502 |
| 16:0 Liss Rhod PE | Avanti Polar Lipids | 810158 |
| 18:1 PS (DOPS) | Avanti Polar Lipids | 840035 |
| 18:1 PI(4,5)P2 | Avanti Polar Lipids | 850155 |
| Cholesterol | Avanti Polar Lipids | 700100 |
| Guanosine triphosphate (GTP) | Jena Bioscience | NU-1012 |
| Guanosine diphosphate (GDP) | Jena Bioscience | NU-1172 |
| Guanosine monophosphate (GMP) | Jena Bioscience | NU-1028 |
| Guanosine 5′-O-[gamma-thio]triphosphate (GTPγS) | Jena Bioscience | NU-410 |
| 5′-Guanylyl imidodiphosphate (GMPPNP) | Jena Bioscience | NU-401 |
| *N*-methylanthraniloyl guanosine-5′-[(β,γ)-imido]triphosphate (mant-GMPPNP) | Jena Bioscience | NU-207 |
| 2′/3′-O-(*N*-methyl-anthraniloyl)-guanosine-5′-diphosphate (mant-GDP) | Jena Bioscience | NU-204 |
| **Oligonucleotides** | | |
| Δ207-213_fw: A GGT GGA GGT AGT CAA AAA GAT GAA ACT TTT AAC C | This study | N/A |
| Δ207-213_rv: GA TCC ACC GCC CAG GGA GTA TGT CAG GTA C | This study | N/A |
| Δ207-216_fw: GGC GGA GGT TCT GAA ACT TTT AAC CTG CCC | This study | N/A |

**Table 1.** (continued)

| Reagent or resource | Source | Identifier |
|---|---|---|
| Δ207-216_rv: AGA ACC TCC GCC CAG GGA GTA TGT CAG GTA C | This study | N/A |
| Δ207-223_fw: GGT TCT CTC TGT ATC AGG AAG TTC TTC | This study | N/A |
| Δ207-223_rv: TCC GCC CAG GGA GTA TGT CAG GTA C | This study | N/A |
| M139D_fw: GGA ACC ATC AAC CAG CAG GCT GAT GAC CAA CTG TAC TAT GTG ACA | This study | N/A |
| M139D_rv: TGT CAC ATA GTA CAG TTG GTC ATC AGC CTG CTG GTT GAT GGT CC | This study | N/A |
| M139E_fw: GGA ACC ATC AAC CAG CAG GCT GAG GAC CAA CTG TAC TAT | This study | N/A |
| M139E_rv: ATA GTA CAG TTG GTC CTC AGC CTG CTG GTT GAT GGT CC | This study | N/A |
| M139R_fw: CC ATC AAC CAG CAG GCT AGG GAC CAA CTG TAC TAT | This study | N/A |
| M139R_rv: ATA GTA CAG TTG GTC CCT AGC CTG CTG GTT GAT GG | This study | N/A |
| D199A_fw: CAA CCC CTC ACA CCA GCT GAG TAC CTG ACA TAC | This study | N/A |
| D199A_rv: GTA TGT CAG GTA CTC AGC TGG TGT GAG GGG TTG | This study | N/A |
| D199K_fw: GGA CAA CCC CTC ACA CCA AAG GAG TAC CTG ACA TAC TCC | This study | N/A |
| D199K_rv: GGA GTA TGT CAG GTA CTC CTT TGG TGT GAG GGG TTG TCC | This study | N/A |
| A318*_fw: G GAG AAC GCA GTC CTG TAA TTG GCC CAG ATA GAG | This study | N/A |
| A318*_rv: CTC TAT CTG GGC CAA TTA CAG GAC TGC GTT CTC C | This study | N/A |

50 mM Tris-HCl pH 7.9, 150 mM NaCl, 5 mM MgCl$_2$, 2 mM DTT according to Dickinson et al (2023). For biochemical studies, non-farnesylated constructs of GBP1 and the isolated LG domains were purified by affinity chromatography (Ni Sepharose HP, Cytiva) and size-exclusion chromatography (Superdex 200 16/600, GE Healthcare). Farnesylated GBP1 constructs were co-expressed with recombinant farnesyltransferase (FTase) from pRSFDuet-1 and purified by affinity chromatography using Ni Sepharose HP, followed by hydrophobic interaction chromatography (Butyl Sepharose HP, GE Healthcare), and size-exclusion chromatography (Superdex 200 16/600, GE Healthcare).

## Protein labeling with fluorescent dye

For FRET studies, the cysteine mutant Q577C was labeled with AlexaFluor488-C5-maleimide dye (AF488, Thermo Fisher Scientific) as recently described (Sistemich and Herrmann, 2020). Briefly, DTT was removed from purified protein by ultrafiltration and buffer exchange to 50 mM Tris-HCl pH 7.4, 150 mM NaCl, 5 mM MgCl$_2$. For in vitro binding assays, GBP1 wild-type and mutants were labeled as described above with AlexaFluor647-C2-maleimide dye (AF647, Thermo Fisher Scientific). Labeling was performed at an equimolar ratio of protein and dye on ice for 10 min. The labeling reaction was stopped by the addition of 2 mM DTT and unreacted dye was removed by ultrafiltration and buffer exchange to 50 mM Tris-HCl pH 7.9, 150 mM NaCl, 5 mM MgCl$_2$. Protein concentration and degree of labeling (DOL) were determined according to Lambert-Beer law at the wavelengths 280 nm (GBP1, $\varepsilon = 45,400\,M^{-1}\,cm^{-1}$), 491 nm (AF488, $\varepsilon = 71,000\,M^{-1}\,cm^{-1}$), and 651 nm (AF647, $\varepsilon = 268,000\,M^{-1}\,cm^{-1}$) and using correction factors for AF488 (0.11) and AF647 (0.03). The DOL of GBP1-Q577C-AF488 was typically between 0.9 and 0.95, the DOL for GBP1-AF647 constructs between 0.1 and 0.15.

## Liposome preparation

Liposomes were prepared from Porcine brain polar lipids (BPL, Avanti Polar Lipids) or Folch fraction I bovine brain lipids (Folch,

Sigma-Aldrich) in glass tubes by drying 40 µl of a 25 mg/ml lipid stock in chloroform under an argon stream. For FRET measurements, BPLs were supplemented with 5 µl of a 1 mg/ml 1,2-dipalmitoyl-sn-glycero-3-phosphoethanolamine N-(lissamine rhodamine B sulfonyl) stock in chloroform (Liss Rhod PE, Avanti Polar Lipids). Residual chloroform was removed under vacuum for 60 min. The homogenous lipid film was rehydrated in 50 mM Tris-HCl pH 7.9, 150 mM NaCl, 5 mM MgCl$_2$ to reach a final lipid concentration of 1 mg/ml. Lipid rehydration was performed in a water bath at 55 °C for 60 min, followed by five freeze–thaw cycles. Liposomes were extruded to 0.1 µm filters using a mini extruder (Avanti Polar Lipids) and stored at 4 °C.

## Liposome co-sedimentation assay

For co-sedimentation assays, 5 µM farnesylated GBP1 was mixed with 0.5 mg/ml extruded liposomes (0.1 µm) and the respective nucleotide (2 mM GTP, 500 µM GTPγS, 500 µM GMPPNP, 200 µM GDP•AlF$_x$, 1 mM GDP) in 40 µl of oligomerization buffer (50 mM Tris-HCl pH 7.9, 150 mM NaCl, 5 mM MgCl$_2$) or oligomerization buffer supplemented with 300 µM AlCl$_3$ and 10 mM NaF. Samples were incubated for 1 min (GTP), 5 min (GTPγS, GMPPNP) or 10 min (apo, GDP•AlF$_x$, GDP) at RT. Liposomes were sedimented via ultracentrifugation for 10 min at 70,000 rpm (TLA-100, Beckman) and 20 °C. Supernatants and pellets were analyzed by SDS-PAGE.

## FRET-based liposome-binding assay

The FRET-based liposome-binding assay was performed with AF488 labeled GBP1-Q577C (donor, D) and liposomes supplemented with 0.5% Liss Rhod PE (acceptor, A) in oligomerization buffer (apo, GTP) or oligomerization buffer supplemented with 300 µM AlCl$_3$ and 10 mM NaF (GDP•AlF$_x$). Liposomes (0.5 mg/ml) were incubated with varying protein concentration in the absence or presence of 2 mM GTP and 200 µM GDP•AlF$_x$, respectively. Measurements were performed in a total volume of 100 µl using a

microplate reader (Spark, Tecan). Emission spectra were recorded for $\lambda_{ex,D} = 495$ nm (7.5 nm bandwidth) before adding nucleotides, and after 1 min (GTP) and 15 min (GDP•AlF$_x$) of incubation a RT. Spectra were corrected for the contributions from direct A excitation upon D excitation. FRET efficiencies were calculated using the ratio between D and A emission intensities ($I_D$ and $I_A$) following D excitation at $\lambda_{em,D} = 520$ nm and $\lambda_{em,A} = 590$ nm, respectively.

## Negative-stain TEM

For negative-stain EM analysis, 10 µM farnesylated GBP1 was reconstituted on 0.5 mg/ml liposomes in oligomerization buffer (50 mM Tris-HCl pH 7.9, 150 mM NaCl, 5 mM MgCl$_2$) supplemented with 2 mM GTP or 200 µM GDP, 300 µM AlCl$_3$ and 10 mM NaF. After 1 min (GTP) or 10 min (GDP•AlF$_x$) of incubation at RT, 1:10 dilutions were applied to glow-discharged carbon-coated copper grids (300 mesh) and stained with 2% (w/v) uranyl acetate. All samples were imaged using a Talos L120C electron microscope (Thermo Fisher Scientific) at 120 kV on a Ceta Detector. For quantification of uniform protein coat formation, the number of gaps per GBP1-coated liposomes was assessed in a blinded experiment. For each construct, 100 micrographs of single liposomes were collected randomly and gaps within the protein coat were counted by seven independent experimenters. Averages were calculated and the $P$ values were derived by an unpaired $t$ test.

## Cryo-EM data collection

Farnesylated GBP1 reconstituted on liposomes (as described above) were supplemented with 5 nm colloidal gold and applied on glow-discharged UltrAuFoil R 2/2 gold grids (Quantifoil Micro Tools), flash-frozen in liquid ethane using a Vitrobot Mark II device (FEI), and stored under liquid nitrogen conditions. A total set of 104 tilt series was imaged on a Titan Krios G3i electron microscope (Thermo Fisher Scientific) operated at 300 kV. Tilt series of $+/-60°$ were acquired at magnification of 42,000× (1.1 Å per pixel in super-resolution mode) using SerialEM with a Hybrid-STA (Sanchez et al, 2020). Frame stacks with an exposure dose of 2.8 e⁻/Å$^2$ for non-zero tilted projection and 14.4 e⁻/Å$^2$ for zero tilted projection were recorded on a Gatan K3 electron detector. Defocus values varied from $-2.0$ µm to $-5.0$ µm.

For polymeric disk formation, 5 µM farnesylated GBP1 was incubated with 200 µM GDP, 300 µM AlCl$_3$ and 10 mM NaF (GDP•AlF$_x$) for 20-30 min, applied on glow-discharged UltrAuFoil R 1.2/1.3 gold grids (Quantifoil Micro Tools), flash-frozen in liquid ethane using a Vitrobot Mark II device (FEI), and stored under liquid nitrogen conditions. In all, 6,483 micrographs were acquired on a Titan Krios G3i electron microscope (Thermo Fisher Scientific, 300 kV) at magnification of 81,000× (0.53 Å per pixel in super-resolution mode) using EPU (Thermo Fisher Scientific). Frame stacks with an exposure dose of 1.165 e⁻/Å$^2$ were recorded on a Gatan K3 electron detector using defocus values between $-0.7$ µm and $-2.0$ µm.

## Image processing

For cryo-ET of GBP1-coated liposomes, raw images output were directly fed to TomoBEAR (Balyschew et al, 2023), a workflow

engine for streamlined processing of cryo-electron tomographic data for subtomogram averaging. Briefly, the TomoBEAR pipeline was described as follows: Raw images were sorted according to different tilt series, MotionCor2 (Zheng et al, 2017) was used for correcting beam-induced motion, IMOD (Kremer et al, 1996) was used for generating tilt stacks, fiducial beads were detected by Dynamo (Castano-Diez et al, 2012), GCTF (Zhang, 2016) was used for defocus estimation, IMOD was then used for fiducial model refinement and tomographic reconstruction.

Coordinates for sub-tomograms were manually generated by DipoleSet models to describe vesicular geometry. Each liposome was handled as a dipole with a center and a radius which could output the coordinate of the membrane surface. Segments were cropped and classified by Dynamo. In brief, preliminary classification was executed at pixel size of 17.6 Å, classes with clear density of membrane and GBP1 protein were selected for classification at pixel size of 4.4 Å. Subsequently, a class with clear GBP1 density was selected for the Dynamo 3D refinement. Resolution and Fourier Shell Correlation curve was computed as described by Scheres and Chen (2012).

For single particle analysis of polymeric disks, dose-fractionated image stacks were subjected to MotionCor2 (Zheng et al, 2017) and CTFFIND4 (Rohou and Grigorieff, 2015) in Relion4.0 (Kimanius et al, 2021). An initial particle set of 101,449 particles (4.24 Å pixel size) was generated by manual and template-based particle picking in Relion4.0 followed by 2D classification rounds for cleaning. Afterwards, 3D classification using an initial reconstruction of a particle subset was performed and best classes were selected (53,452 particles). Following several rounds of 3D classification using a manual created low-pass filtered map of one planar disk lacking GED domains, a final set of 15,952 particles allowed for an initial low-resolution 3D reconstruction (~37 Å) of stacked polymeric disks without applying symmetry, displaying the general composition of its assembly.

## Circular dichroism (CD) measurements

GBP1 constructs were dialyzed against 10 mM potassium phosphate pH 7.8, 150 mM NaF and CD spectra were recorded from 260 nm to 190 nm using a Jasco J-720 spectropolarimeter at 20 °C. Samples were scanned at 2.5 µM in quartz cuvettes with a pathlength of 0.1 cm. Averages of ten buffer-subtracted spectra were analyzed in CDNN 2.1 (Bohm et al, 1992).

## Polymer crosslinking assay

For crosslinking, a buffer exchange of non-farnesylated and farnesylated GBP1 stored in Tris-HCl to crosslinking buffer (50 mM HEPES, 150 mM NaCl, 5 mM MgCl$_2$) was performed by ultrafiltration. Crosslinking was carried out in a 96-well plate using 20 µM protein. For each construct, non-farnesylated and farnesylated protein were incubated without any nucleotide (apo) or with 200 µM GDP, 300 µM AlCl$_3$ and 10 mM NaF (GDP•AlF$_x$) for 15 min. Samples were mixed with the homobifunctional amine-to-amine crosslinker bis(sulfosuccinimidyl)suberate (BS$^3$, Thermo Fisher Scientific) at 20-times molar access and incubated for 30 min at RT. The crosslinking reaction was stopped by adding 0.5 M Tris-HCl pH 7.9 and samples were further diluted in SDS

sample buffer and analyzed by SDS-PAGE loading 1.0 μg protein per well.

## Light scattering-based polymerization assay

Polymerization of farnesylated GBP1 was monitored by measuring the apparent changes in absorbance as result of changes in light scattering over time. Measurements were performed at a protein concentration of 10 μM (GTP) or 20 μM (GDP•AlF$_x$) in oligomer-ization buffer (50 mM Tris-HCl pH 7.9, 150 mM NaCl, 5 mM MgCl$_2$) at a total volume of 100 μl. Polymerization was initiated by adding 1 mM GTP or 200 μM GDP, 300 μM AlCl$_3$ and 10 mM NaF, respectively. Apparent changes in absorbance were measured using a microplate reader (Infinite 200, Tecan) at a wavelength of 350 nm (9 nm bandwidth) over 40 min. Signals were corrected for the contribution of protein alone.

## Nucleotide-binding assay

Nucleotide binding was analyzed using the non-hydrolysable GTP analog N-methylanthraniloyl guanosine-5'-[(β,γ)-imido] triphosphate (mant-GMPPNP, Jena Bioscience) or mant-GDP (Jena Biosciences). Varying protein concentrations of non-farnesylated GBP1 or isolated LG domain ranging from 0.02 μM to 40 μM were prepared in a 96-well plate in 50 mM Tris-HCl pH 7.9, 150 mM NaCl, 5 mM MgCl$_2$. After adding 0.5 μM mant-GMPPNP or mant-GDP to each well, fluorescence was excited at $\lambda_{ex}$ = 355 nm (10 nm bandwidth) and fluorescence emission was detected at $\lambda_{em}$ = 448 nm (10 nm bandwidth) in a microplate reader (Spark, Tecan). Fluorescence was corrected for mant-nucleotides only. For each construct, averages of three independent measurements were plotted against the protein concentration. Equilibrium dissociation constants $K_d$ were calculated by fitting a quadratic equation (Eq. 1) to data according to Kunzelmann et al (2005).

$$F = F_{min} + (F_{max} - F_{min})\frac{A_0 + B_0 + K_d - \sqrt{(A_0 + B_0 + K_d)^2 - 4A_0B_0}}{2B_0}$$

(1)

## GTP and GDP hydrolysis assays

The catalytic GTPase activity of non-farnesylated GBP1 and the catalytic GDPase activity of isolated LG domains were analyzed by mixing varying protein concentrations with 1 mM GTP or GDP at 37 °C in 50 mM Tris-HCl pH 7.9, 150 mM NaCl, 5 mM MgCl$_2$. Aliquots were taken at defined time points, diluted in HPLC buffer (100 mM potassium phosphate pH 6.5, 10 mM tetrabutylammo-nium bromide, 7.5% (v/v) acetonitrile), heat-inactivated at 80 °C for 1 min, and either analyzed directly or stored at −20 °C. All reactions were performed in duplicates. Nucleotides were separated by reversed-phase HPLC (Hypersil ODS C18, Thermo Fisher Scien-tific) and nucleotide composition was analyzed by integration of the nucleotide absorbance at 254 nm. Initial rates of GTP or GDP hydrolysis were obtained from linear regression. Specific activities were calculated by dividing the initial rates by protein concentra-tion. Apparent $K_d$ values for dimer formation and maximum

catalytic activities ($k_{max}$) were calculated by fitting a quadratic equation (Eq. 2) to data according Praefcke et al (1999).

$$\text{Specific activity} = k_{max}\frac{A_0 + \frac{K_d}{2} - \sqrt{\left(A_0 + \frac{K_d}{2}\right)^2 - A_0^2}}{A_0}$$

(2)

## Analytical size-exclusion chromatography

Dimer formation of non-farnesylated GBP1 and isolated LG domains were analyzed by analytical size-exclusion chromatogra-phy (Superdex 200 5/150, GE Healthcare) as described (Ince et al, 2017). Samples of 20 μM protein without nucleotide (apo) and with the addition of 200 μM GDP or GMP, 300 μM AlCl$_3$ and 10 mM NaF (GDP•AlF$_x$ or GMP•AlF$_x$) were incubated at RT for at least 10 min before subjecting to analytical size-exclusion chromatography in 50 mM Tris-HCl pH 7.9, 150 mM NaCl, 5 mM MgCl$_2$.

## In vitro binding assay

Binding of GBP1 to bacteria was analyzed as described previously (Dickinson et al, 2023; Kutsch et al, 2020). Briefly, an overnight culture of pathogenic E. coli (DSM1103) expressing eGFP was diluted 1:30 and grown at 37 °C under shaking at 140 rpm in 5 ml LB. After 2 hours, 3 ml bacterial culture was pelleted, washed with 1× PBS, and fixed in 4% formaldehyde in PBS, pH 7.4 for 20 min. Formaldehyde-fixed bacteria were washed twice with 1× PBS. For the in vitro binding assays, bacteria were diluted in 50 mM Tris-HCl pH 7.9, 150 mM NaCl, 5 mM MgCl$_2$, 50 μM BSA and shortly centrifuged at 500 × g onto the cover slide of a glass bottom 10 mm microwell dish. Wild-type and mutant GBP1-AF647 were diluted in 50 mM Tris-HCl pH 7.9, 150 mM NaCl, 5 mM MgCl$_2$, 50 μM BSA, mixed with GTP, and the mixture was added to and gently mixed with bacteria at $t$ = 0 min (final concentrations: $10^5$–$3 \times 10^6$ bacteria/ml, 10 μM GBP1, and 2 mM GTP). Time-lapse confocal microscopy imaging was used to visualize GBP1 binding to bacteria over time. Images were collected every 1.5 min and after 30 min, different field of views were imaged for quantification. Imaging was performed on an inverted Zeiss 880 Airyscan Laser Scanning Confocal Microscope using a Zeiss Plan-Apochromat ×63/1.4 oil DIC M27 objective. All images were processed with Fiji. Quantitative analysis of high content confocal microscopy images was done using the machine learning-based image analysis platform HRMAn2.0 (Fisch et al, 2021, 2019b).

# Data availability

Single particle cryo-EM and subtomogram averaging structures have been deposited in the Electron Microscopy Data Bank (EMDB) under accession codes EMD-18698 and EMD-18806, respectively. Coordinates for the membrane-bound GBP1 model have been submitted to the Protein Data Bank (PDB) under accession code 8R1A.

## Peer review information

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

## Acknowledgements

The authors thank Dr. Thiemo Sprink, Metaxia Stavroulaki, and Dr. Christoph Diebolder from the core facility for cryo-EM at Charité Universitätsmedizin Berlin for cryo-EM grid preparation and data collection. The Core Facility was supported by the German Research Foundation through grant No. INST 335/588-1 FUGG. We also thank Antonia Nowicki and Varvara Plotnikova for support in protein production; Dr. Julia Smirnova for assistance in single particle analysis, Dr. Tobias Bock-Bierbaum, Dr. Katja Fälber, Evangelia Nathanail, Larissa Obst, Dr. Yvette Roske and Elena Vázquez for their contribution in the blinded quantification experiment; and Dr. Gerrit Praefcke for sharing the pQE-80L-GBP1 and pRSFDuet-1-FTase plasmids. We thank the Center for Advanced Imaging (CAi) at Heinrich-Heine-University Düsseldorf for providing access to the Zeiss LSM 880 Airyscan Fast (Funding for instrumentation: DFG-INST 208/746-1 FU) and Dr. Oxana Krylova and Heike Nikolenko from the Biophysics Unit at Leibniz-Forschungsinstitut für Molekulare Pharmakologie (FMP) for providing access to the Jasco J-720 CD spectropolarimeter and assistance in performing the experiments. This work was supported by the NRW-Rückkehrprogramm of the State of North Rhine-Westphalia (to M Kutsch) and the iNAMES-MDC-Weizmann Helmholtz International Research School (HIRS) for Imaging and Data Science from the NAno to the MESo (to MW). We thank Dr. Hagen Hofmann (Weizmann Institute of Science) for helpful suggestions and constructive discussion.

## Author contributions

**Marius Weismehl**: Conceptualization; Formal analysis; Investigation; Visualization; Writing—original draft; Writing—review and editing; MW generated and purified all constructs, designed and performed the experiments, analyzed the data, processed single particle cryo-EM data, and contributed to the cryo-ET image analysis. **Xiaofeng Chu**: Formal analysis; Investigation; Visualization; XC processed cryo-ET data and performed image analysis. **Miriam Kutsch**: Formal analysis; Investigation; Visualization; Writing—review and editing; M Kutsch performed and analyzed the in vitro binding assay. **Paul Lauterjung**: Formal analysis; Investigation; PL purified protein for cryo-ET and supported in initial protein-coated liposome preparation. **Christian Herrmann**: Formal analysis; Supervision; Funding acquisition. **Misha Kudryashev**: Formal analysis; Supervision; Funding acquisition. **Oliver Daumke**: Conceptualization; Formal analysis; Supervision; Funding acquisition; Writing—original draft; Project administration; Writing—review and editing.

## Funding

## Disclosure and competing interests statement

The authors declare no competing interests.

# Expanded View Figures

**Figure EV1. Subtomogram averaging pipeline of membrane-bound GBP1.**

(**A**) Cropped segments from GBP1 tomograms are displayed by "dynamo_gallery." (**B**) Two steps of 3D classification were implemented in pixel size of 17.6 Å and 4.4 Å. Boxes of the dashed line denote the selected classes for next steps. (**C**) 3D refinement result of the selected class and gold-standard FSC curve. Map was low-passed to 25 Å.

▶

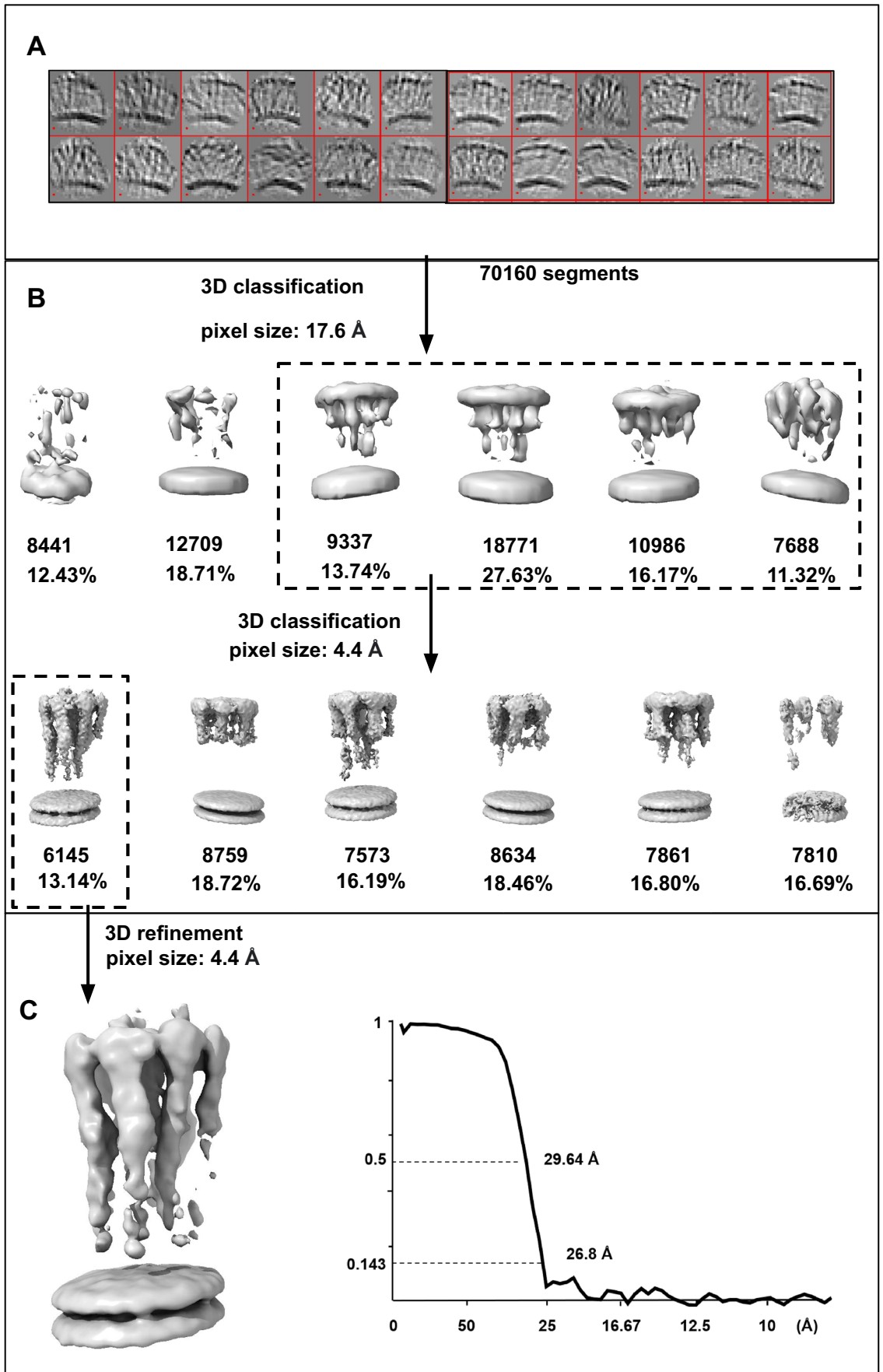

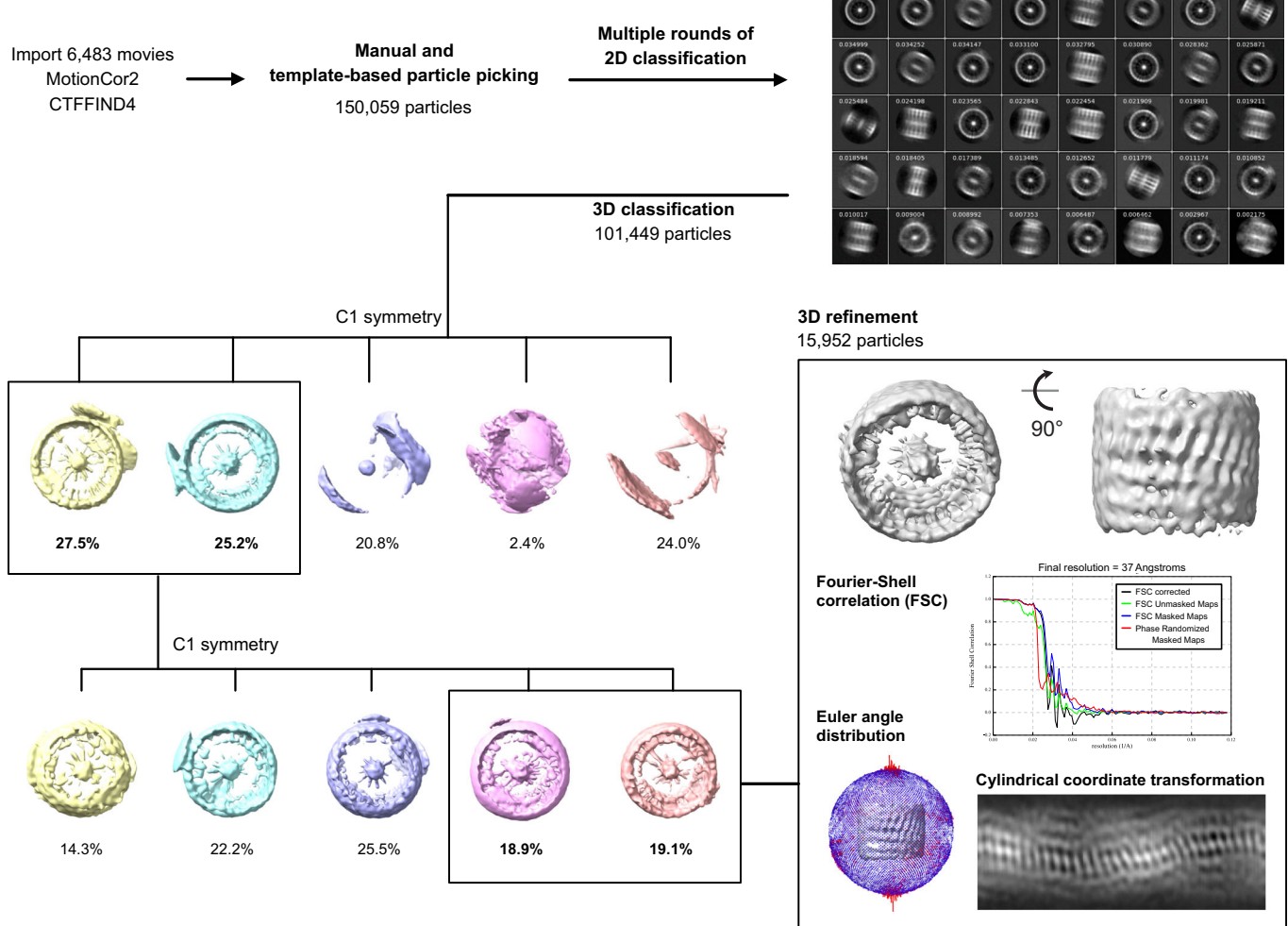

**Figure EV2. Image processing workflow for polymeric GBP1 disks.**

Boxes denote selected 3D classes for next steps. 3D refinement result is shown together with gold-standard FSC curves, Euler angle distribution of particles contributing to the final reconstruction, and cylindrical coordinate transformation visualizing the surface of the 3D reconstruction at the heights of the peripheral LG domains.

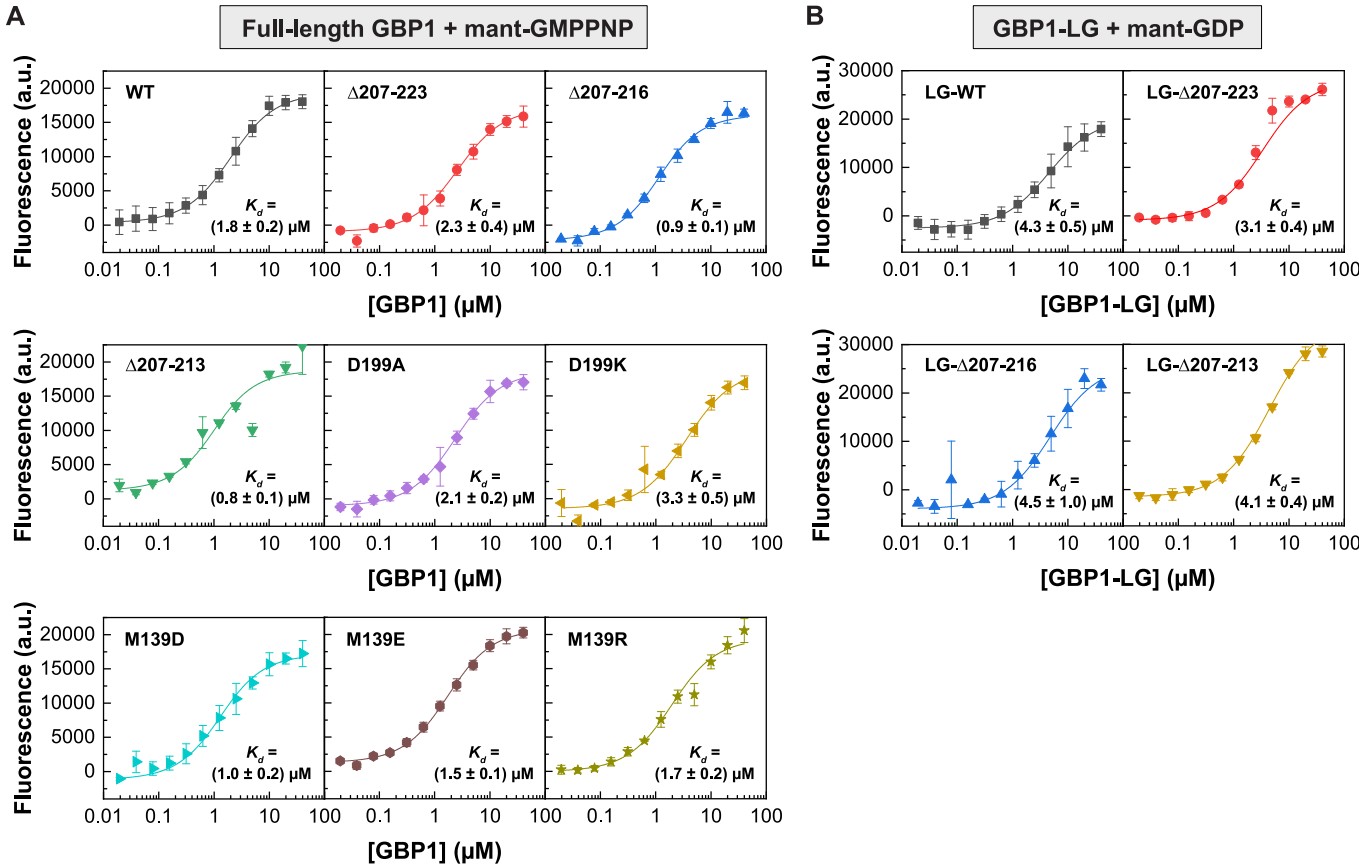

**Figure EV3. Nucleotide binding.**

(A) Fluorescence of mant-GMPPNP (0.5 μM) at varying GBP1 concentration for indicated full-length constructs. Data points are averages from three independent experiments and are represented by mean ± SD. Equilibrium dissociation constants $K_d$ were calculated by fitting a quadratic equation to data using Eq. 1. (B) Fluorescence of mant-GDP (0.5 μM) at varying GBP1 concentration for indicated isolated LG domain constructs. Data points are averages from three independent experiments and are represented by mean ± SD. Equilibrium dissociation constants $K_d$ were calculated by fitting a quadratic equation to data using Eq. 1. Source data are available online for this figure.

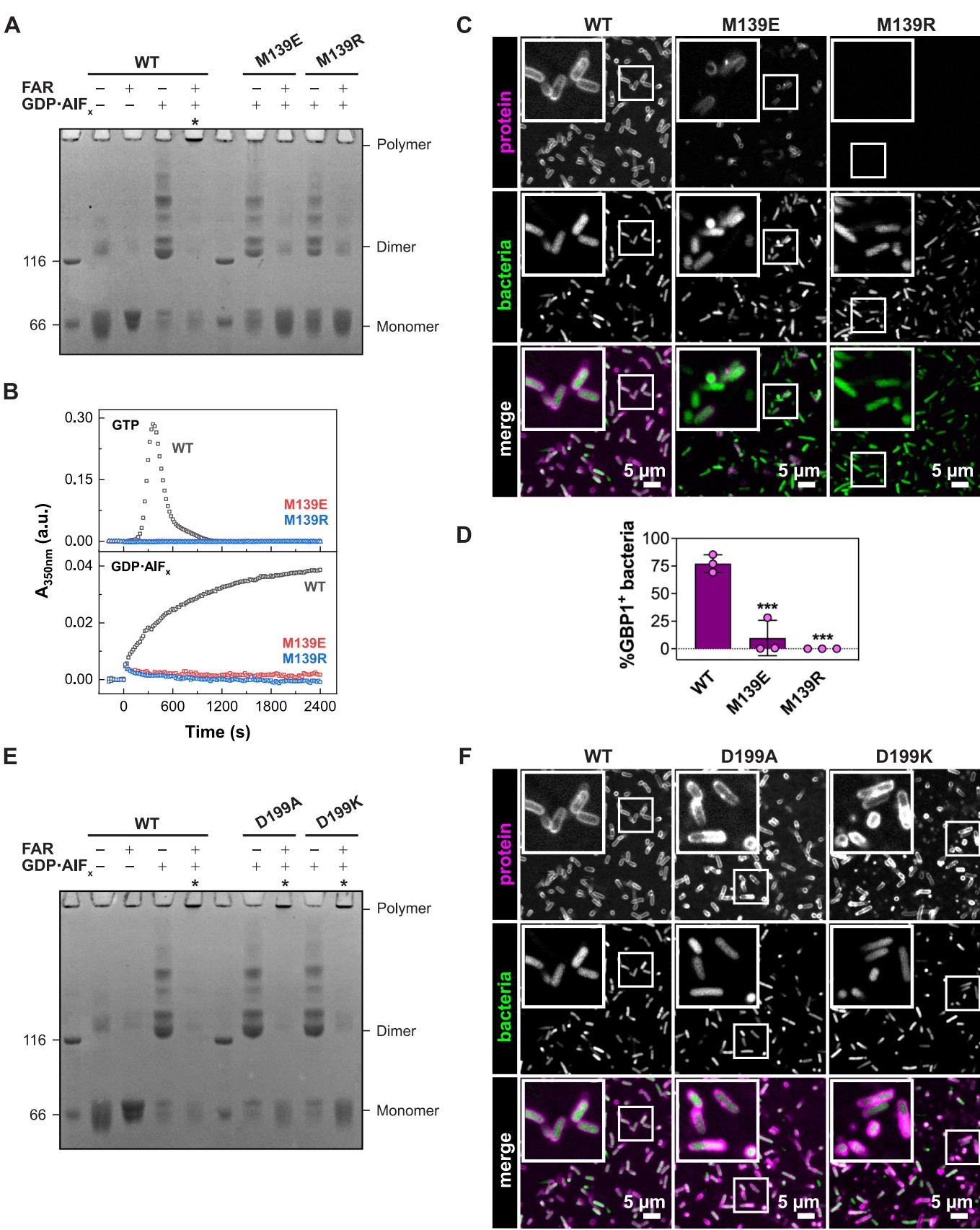

◄  **Figure EV4.  Oligomerization of dimerization-capable M139 and pivot point mutants.**

(A) Crosslinking assay of M139 mutants. The respective oligomeric states based on molecular weight are indicated. Asterisks indicate polymerization. FAR farnesylated. (B) Light scattering-based polymerization assay of M139 mutants. Polymerization is induced by GTP (top) and GDP•AlF$_x$ (bottom). WT data is the same as shown in Fig. 2D and provided here for comparison. (C) Confocal microscopy images of wild-type GBP1 and M139 mutants targeting pathogenic *E. coli*, as described in the legend of Fig. 3F. (D) Quantification of GBP1-targeted bacteria in in vitro binding assay, as described in the legend of Fig. 3G. (E) Cross-linking assay of D199 mutants. The respective oligomeric states based on molecular weight are indicated. Asterisks indicate successful polymerization. FAR farnesylated. (F) Confocal microscopy images of wild-type GBP1 and D199 mutants targeting pathogenic *E. coli*, as described in Fig. 3F. Source data are available online for this figure.

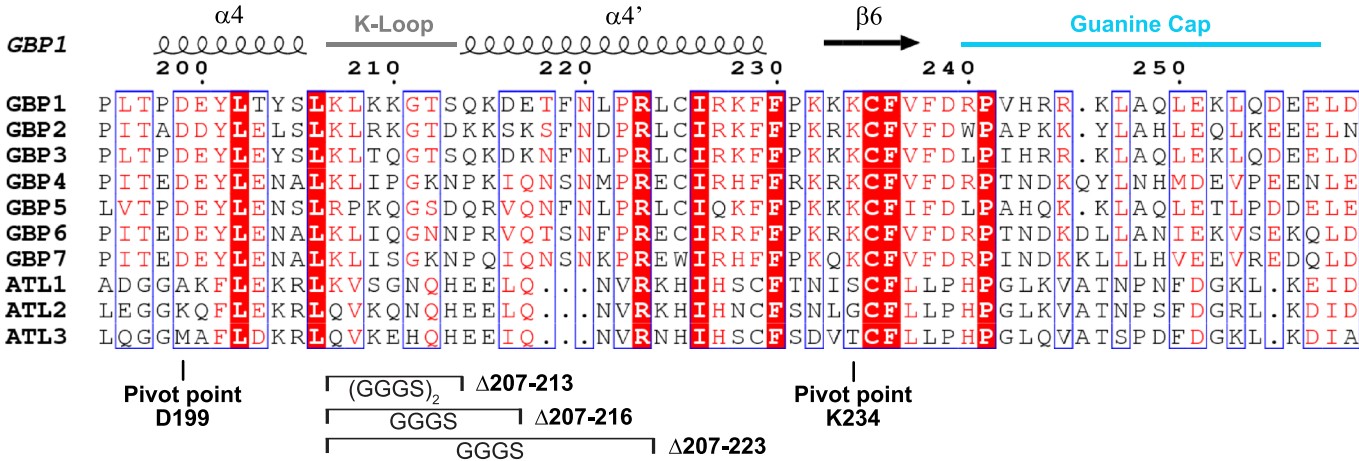

**Figure EV5. Sequence alignment of the intramolecular lever.**

Structural elements of PDB 1dg3 (GBP1) are shown. The pivot point residues and the generated helix α4' variants for GBP1 are marked. GBP human guanylate-binding protein, ATL human atlastin.

