## [Peer Review File · The EMBO Journal]

Structural insights into the activation mechanism of antimicrobial GBP1

Marius Weismehl, Xiaofeng Chu, Miriam Kutsch, Paul Lauterjung, Christian Herrmann, Misha Kudryashev, and Oliver Daumke
DOI: 10.15252/emboj.2023115158

Corresponding author(s): Oliver Daumke (oliver.daumke@mdc-berlin.de)

Review Timeline:

Submission Date:	31st Jul 23
Editorial Decision:	24th Aug 23
Revision Received:	14th Nov 23
Editorial Decision:	11th Dec 23
Revision Received:	11th Dec 23
Accepted:	14th Dec 23

Editor: Ioannis Papaioannou

Transaction Report:

Dear Dr. Daumke,

Thank you for submitting your manuscript for consideration by the EMBO Journal. It has now been seen by three experts in the field, and we have received the full set of their reports, which are included below.

As you will see, all referees are positive about the study. They explain that the presented findings are interesting and significant, and that the advance provided is considerable. However, they also point out a few issues that require clarification, and they further provide a number of suggestions for the improvement of the study and the manuscript.

Given the referees' positive comments and recommendations, I would like to invite you to submit a revised version of your manuscript, addressing the comments of all three reviewers. I should add that it is EMBO Journal policy to allow only a single round of major revision, and acceptance of your manuscript will therefore depend on the completeness of your responses in this revised version. If you have any questions or comments, we can also discuss the revisions in a video chat, if you like.

We generally allow three months as standard revision time (23rd November 2023). As a matter of policy, competing manuscripts published during this period will not negatively impact our assessment of the conceptual advance presented by your study. However, we request that you contact us as soon as possible upon publication of any related work, to discuss how to proceed. Should you foresee a problem in meeting this three-month deadline, please let us know in advance and we may be able to grant an extension.

Thank you for the opportunity to consider your work for publication in the EMBO Journal. I look forward to your revision.

Yours sincerely,

Instructions for preparing your revised manuscript

1. When you are ready to submit the revision, please upload:

- A Word file of the manuscript text (including legends of main Figures, EV Figures and Tables). Please make sure that changes are highlighted (or "tracked") to be clearly visible.

- Individual production-quality figure files (one file per figure). When assembling your figures, please refer to our figure preparation guidelines in order to ensure proper formatting and readability in print as well as on screen:

If the data shown in a figure are obtained from n {less than or equal to} 2, please use scatter plots showing the individual data points.

i. the name of the statistical test used to generate error bars and P values

ii. the number (n) of independent experiments (please specify technical or biological replicates) underlying each data point (discussion of statistical methodology can be reported in the Materials and Methods section, but figure legends should contain a basic description of n , P , and the test applied)

iii. the nature of the bars and error bars (s.d., s.e.m.).

- A point-by-point response to the referees' comments, with a detailed description of the changes made (as a word file). All referees' concerns must be fully addressed and their suggestions taken on board. When preparing your letter of response to the referees' comments, please bear in mind that this will form part of the Review Process File and will therefore be available online to the community. Please note that you have the possibility to opt out of the transparent process at any stage prior to publication by letting the editorial office know (contact@embojournal.org); if you do opt out, the Review Process File link will point to the following statement: "No Review Process File is available with this article, as the authors have chosen not to make the review process public in this case.". For more details on our Transparent Editorial Process, please visit our website: <https://www.embopress.org/page/journal/14602075/authorguide#transparentprocess>

- Expanded View (EV) files (replacing Supplementary Information) that are collapsible/expandable online. A maximum of 5 EV

Figures can be typeset. EV Figures should be cited as "Figure EV1, Figure EV2" etc. in the text, and their respective legends should be included in the manuscript file after the legends of regular figures. See detailed instructions regarding Expanded View files here:

- For the figures that you do NOT wish to display as Expanded View figures, they should be bundled together with their legends in a single PDF file called "Appendix", which should start with a short Table of Contents (including page numbers). Appendix figures should be referred to in the main text as: "Appendix Figure S1, Appendix Figure S2" etc. Please see detailed instructions here: <https://www.embopress.org/page/journal/14602075/authorguide#expandedview>

- A complete author checklist, which you can download from our author guidelines (<https://www.embopress.org/page/journal/14602075/authorguide>). Please note that the checklist will also be part of the Review Process File.

2. Please note that no statistics should be calculated if $n=2$.

3. Before submitting your revision, primary datasets (and computer code, where appropriate) produced in this study need to be deposited in appropriate public databases (see <https://www.embopress.org/page/journal/14602075/authorguide#dataavailability>). Specifically, we would kindly ask you to provide public access to the following datasets/data:

- Single-particle cryo-EM and subtomogram averaging structures.

The accession numbers and database should be listed in a formal "Data availability" section (placed after Materials and Methods) that follows the model below (see also

<https://www.embopress.org/page/journal/14602075/authorguide#dataavailability>):

Data availability

- RNA-seq data: Gene Expression Omnibus GSE46843 (<https://www.ncbi.nlm.nih.gov/geo/query/acc.cgi?acc=GSE46843>)
- [data type]: [name of the resource] [accession number/identifier/doi] ([URL or identifiers.org/DATABASE:ACCESSION])

*** Note: all links should resolve to a page where the data can be accessed. ***

*** Note: the Data Availability Section is restricted to new primary data that are part of this study. ***

4. Please check that the title and the abstract of the manuscript are brief, yet explicit, even to non-specialists. The length of the title should not exceed 100 characters (including spaces), and the abstract should be a single paragraph not exceeding 175 words.

5. Please also note our reference format: <https://www.embopress.org/page/journal/14602075/authorguide#referencesformat>.

7. Please remember: digital image enhancement is acceptable practice, as long as it accurately represents the original data and conforms to community standards. If a figure has been subjected to significant electronic manipulation, this must be noted in the figure legend or in the "Materials and Methods" section. The editors reserve the right to request original versions of figures and the original images that were used to assemble the figure.

8. Our journal encourages inclusion of data citations in the reference list to directly cite datasets that were obtained from public databases. Data citations in the article text are distinct from normal bibliographical citations and should directly link to the database records from which the data can be accessed. In the main text, data citations are formatted as follows: "Data ref: Smith et al, 2001" or "Data ref: NCBI Sequence Read Archive PRJNA342805, 2017". In the Reference list, data citations must be labeled with "[DATASET]". A data reference must provide the database name, accession number/identifiers, and a resolvable link to the landing page from which the data can be accessed at the end of the reference. Further instructions are available at: <https://www.embopress.org/page/journal/14602075/authorguide#referencesformat>.

9. We request authors to consider both actual and perceived competing interests. Please review our policy (<https://www.embopress.org/page/journal/14602075/authorguide#conflictofinterest>) and update your competing interests statement if necessary. Please name this section 'Disclosure and competing interests statement' and place it after the

Acknowledgements section.

10. Please note that all corresponding authors are required to provide an ORCID ID upon submission of a revised manuscript (<https://orcid.org/>). Please find instructions on how to link your ORCID ID to your account in our manuscript tracking system in our Author guidelines (<https://www.embopress.org/page/journal/14602075/authorguide#authorshipguidelines>).

11. We use CRediT to specify the contributions of each author in the journal submission system. CRediT replaces the author contribution section, which should be removed from the manuscript. Please use the free text box to provide more detailed descriptions. See also guide to authors: <https://www.embopress.org/page/journal/14602075/authorguide#authorshipguidelines>.

13. We would also welcome the submission of cover suggestions or motifs to be used by our Graphics Illustrator in designing a cover.

14. Please use the link below to submit your revision:
<https://emboj.msubmit.net/cgi-bin/main.plex>

Yours sincerely,

Referee #1:

This manuscript presents a structural analysis of the guanylate-binding protein GBP1 using cryoEM and cryoET with sub-tomogram averaging of membrane assembled oligomers, combined with structural and kinetic analyses of the GTP-hydrolysis steps in GBP1 assembly and activity on bacterial infection in cells. The major findings were that GBP1 can assemble into discs and stacked disc assemblies with a highly extended GBP1 conformation, as well as form an extended membrane tethered polymer. Mutagenesis, biochemical and cellular studies point to a nucleotide-dependent series of steps leading to structural formation of dimers, oligomers, and membrane-assembled coatamer structures. The clever use of specific GBP1 mutants allowed for the testing of hypotheses regarding GBP1 oligomer formation and GTP/GDP hydrolysis. I would note that there is a manuscript on Biorxiv by Kuhm et al., that presents related findings (referenced in this manuscript), but in my opinion this adds to the confidence in the findings rather than detracting from any novelty.

I thought the paper was well written, the figures of high quality, the methods seem sound, and it provides interesting new insights into activity of GBP1 and the broader GBP family. For context I have expertise in the structural biology of membrane-interacting proteins, but I am not a specific expert on GBP proteins or cryoET methods. I don't have any significant queries or critiques, and only a few minor questions.

Queries.

1. Fig 1A. (line 825). Is there a reference for Fig. 1A?
2. Fig 4F. It looks like the WT light-scattering curve shown here is the same curve as in Fig 2D. If so it suggests that the data is not an independent replicate, which is not a problem but should be stated clearly (e.g. WT data is the same as shown in Fig. 2D and provided here for comparison).
3. It would be good to provide a little more detail for the in vitro farnesylation reactions including protein and enzyme concentrations and enzyme source.

Referee #2:

In this study, Weismehl et al. address the coupling of nucleotide state of human GBP1 to dimerization and oligomerisation, in the context of both soluble oligomers and membrane-bound coatamers on liposomes and pathogenic *E. coli*.

Using comparison of previous structural work alongside their own low-resolution cryo-ET and cryo-EM reconstructions of the

GBP1 membrane-independent oligomers and coatomer, Weismehl et al. design mutants to demonstrate the role of $\alpha 4$ in the LG (GTPase) domain in oligomerisation, and probe how nucleotide state changes influence oligomerisation and membrane binding via coordinated movements within the LG domain. This work is extended to the context of coat formation on pathogenic *E. coli*. This is a significant and important addition to the field and merits publication, given the exciting mechanistic advances this work provides including the remarkable and unexpected extension of the GED.

However, there are a few elements that require clarification:

Major Points

- Can the authors please clarify that the 2D class averages of the soluble oligomer are not helical particles. This can be done by analysing the FFT for presence of layer lines on the clearest 2D classes. The reconstruction in Figure S6 looks like it may also have helical symmetry although it is hard to tell at this level. It may be that some classes contain stacks, but some appear offset, suggesting at least some particles have helical symmetry. If these are short helical filaments, they can also have rotational symmetry, which could appear like stacked planar rings in the side views.
- Could the authors please clarify the mutations in Figure 2B. If my understanding is correct, why has the 207-213 region been replaced with two GGGS repeats, and the 207-216/223 region only been replaced with a single GGGS? On this point, although the EM suggests the overall architecture is maintained in the mutants, the authors should confirm this for example, are gel filtration profiles similar, so that we are certain the localised effects of $\alpha 4$ helix mutations are responsible for the changes in oligomerisation efficiency, and not structural changes induced in the protein elsewhere. Although the authors attempt to explain the increased GTPase activity of the 207-213 mutant, there is limited discussion of why $\alpha 4$ mutants cannot carry out the second hydrolysis step. This should be explored further.
- Could the authors clarify their conclusion at line 324, suggesting the coordinated movement couples nucleotide hydrolysis to oligomerisation. If pivot point mutants can encapsulate *E. coli* efficiently, and oligomerisation isn't affected, how does this support the pivot point facilitating a coordinated movement to allow oligomerisation? The authors should provide evidence for their explanation that LPS of *E. coli* is responsible for stabilising oligomerisation, in a mutant that should have dramatically reduced oligomerisation.

Minor points

- In line 146 it is suggested that the MD and GED domains don't contain oligomerisation interfaces. With the map at this resolution, it is hard to rule out weaker contacts between these domains, especially at the GED closer to the membrane. Flexibility within the oligomer may mean these contacts have been averaged out, especially given the LG domains are likely to have more extensive contacts and a more rigid lattice. It is recommended that the authors clarify in the text that contacts in these regions could have been lost through averaging.
- Could the difference between the BPL and Folch Lipids be discussed? Is there an obvious reason membrane-independent oligomerisation is favoured for the Folch lipids?
- The cross-linked oligomer band for the 207-223 mutant in Figure 2C is described as "completely abolished". However, it appears there is still a faint band at the top of the gel. The authors should acknowledge this.
- Although not in the scope of this manuscript, this work would be greatly strengthened by tomograms of polymers bound to *E. coli*. This would confirm the conformation they are proposing may have a more physiological role.

Referee #3:

In this manuscript, Weismehl and colleagues explored the molecular mechanism underlying the oligomerization process of human GBP1. Combining cryo-EM, cryo-ET, and many biochemical approaches, the authors convincingly identified critical interactions between the $\alpha 3$ and $\alpha 4'$ helices of human GBP1 that regulate its oligomeric state. This is a beautiful piece of work. I have little to criticize and fully support publication on EMBO J.

Minor concerns:

1. What is the lipid-specificity of GBP1? The authors showed in Fig. 1D that GBP1 preferentially forms lipid-independent oligomers in the presence of Folch lipids, while coats BPL. What causes this difference? Is it related to the similarity with bacterial membrane?
2. In Fig. 2A, it would be more helpful to label the shift of the $\alpha 4'$ helix in a more obvious way - it took me a while to identify the mentioned shift.
3. In Fig. 2B, does the length of the GS-linker match the length of the substituted residues, especially for d207-216 and d207-223 mutants?

We would like to thank all three referees for their supporting and positive comments. Please find below our detailed point-by-point response to your concerns.

Referee #1:

This manuscript presents a structural analysis of the guanylate-binding protein GBP1 using cryoEM and cryoET with sub-tomogram averaging of membrane assembled oligomers, combined with structural and kinetic analyses of the GTP-hydrolysis steps in GBP1 assembly and activity on bacterial infection in cells. The major findings were that GBP1 can assemble into discs and stacked disc assemblies with a highly extended GBP1 conformation, as well as form an extended membrane tethered polymer. Mutagenesis, biochemical and cellular studies point to a nucleotide-dependent series of steps leading to structural formation of dimers, oligomers, and membrane-assembled coatamer structures. The clever use of specific GBP1 mutants allowed for the testing of hypotheses regarding GBP1 oligomer formation and GTP/GDP hydrolysis. I would note that there is a manuscript on Biorxiv by Kuhm et al., that presents related findings (referenced in this manuscript), but in my opinion this adds to the confidence in the findings rather than detracting from any novelty.

I thought the paper was well written, the figures of high quality, the methods seem sound, and it provides interesting new insights into activity of GBP1 and the broader GBP family. For context I have expertise in the structural biology of membrane-interacting proteins, but I am not a specific expert on GBP proteins or cryoET methods. I don't have any significant queries or critiques, and only a few minor questions.

Thank you very much for the positive evaluation!

Queries.

1. Fig 1A. (line 825). Is there a reference for Fig. 1A?

We prepared **Fig 1A** with the published crystal structure (PDB 6k1z) by Ji *et al* (2019). We have included this reference in the figure legend and the respective text passage (**line 73/74**).

2. Fig 4F. It looks like the WT light-scattering curve shown here is the same curve as in Fig 2D. If so it suggests that the data is not an independent replicate, which is not a problem but should be stated clearly (e.g. WT data is the same as shown in Fig. 2D and provided here for comparison).

Thank you for pointing this out. We addressed this point as suggested. We have also added this statement to the light scattering-based polymerization assay in **Fig EV4B**.

3. It would be good to provide a little more detail for the in vitro farnesylation reactions including protein and enzyme concentrations and enzyme source.

We added the requested details to the method section.

Referee #2:

In this study, Weismehl et al. address the coupling of nucleotide state of human GBP1 to dimerization and oligomerisation, in the context of both soluble oligomers and membrane-bound coatomers on liposomes and pathogenic *E. coli*.

Using comparison of previous structural work alongside their own low-resolution cryo-ET and cryo-EM reconstructions of the GBP1 membrane-independent oligomers and coatomer, Weismehl et al. design mutants to demonstrate the role of $\alpha 4$ in the LG (GTPase) domain in oligomerisation, and probe how nucleotide state changes influence oligomerisation and membrane binding via coordinated movements within the LG domain. This work is extended to the context of coat

formation on pathogenic E. coli. This is a significant and important addition to the field and merits publication, given the exciting mechanistic advances this work provides including the remarkable and unexpected extension of the GED.

Thank you very much.

However, there are a few elements that require clarification:

Major Points

- Can the authors please clarify that the 2D class averages of the soluble oligomer are not helical particles. This can be done by analysing the FFT for presence of layer lines on the clearest 2D classes. The reconstruction in Figure S6 looks like it may also have helical symmetry although it is hard to tell at this level. It may be that some classes contain stacks, but some appear offset, suggesting at least some particles have helical symmetry. If these are short helical filaments, they can also have rotational symmetry, which could appear like stacked planar rings in the side views.

To address this concern, we first attempted to analyze the 2D classes for the presence of helical patterns, which we, however, could not convincingly detect (**Fig R1**).

Figure R1. Analysis of 2D classes of the GBP1 soluble oligomer. 2D classes used from Fig 1J, representing the clearest classes.

Next, we classified the particles into 4 classes and performed a cylindrical coordinate transformation of our final reconstruction (original) and of the obtained classes (**Fig R2**). The cylindrical transformation visualizes the surface of the 3D reconstruction, which we display at the heights of the peripheral LG domains of the GBP1 polymer. Both in the original reconstruction as well as in the four classes, major distortions are apparent. A helical assembly would correspond to a regular diagonal pattern – from one of the lower corners to a top corner. However, we do not see such a regular pattern of rise that indicates helical assembly in any of the classes, although we cannot exclude that a minor subpopulation of particles is helical. We have revised the corresponding text passage in the result section (**lines 176-182**). Our reconstruction of short soluble oligomers rather suggests a stacking mechanism, supported by the presence of single “planar” disks in our EM data.

We added the cylindrical transformation to the SPA workflow of **Fig EV2**.

Figure R2. Cylindric coordinate transformation of polymeric GBP1 disks. Protein in white. Displayed at the heights of the peripheral LG domains.

- Could the authors please clarify the mutations in Figure 2B. If my understanding is correct, why has the 207-213 region been replaced with two GGGS repeats, and the 207-216/223 region only been replaced with a single GGGS?

For all $\alpha 4'$ variants, we replaced the indicated regions (e.g. amino acids 207 to 223 in the $\Delta 207-223$ variant) with a GGGS-linker. For clarity, we predicted the structure of the $\alpha 4'$ variants using AlphaFold2 (Jumper *et al*, 2021; Mirdita *et al*, 2022) and displayed the relevant regions in **Fig R3**. As illustrated, two GGGS repeats are necessary to bridge the greater distance between amino acids 206 and 214 in the $\Delta 207-213$ variant as compared to the $\Delta 207-216/223$ variants.

Figure R3. Design of helix $\alpha 4'$ variants. *Left:* helices $\alpha 4$, $\alpha 4'$ and the K-Loop of wild-type GBP1 in the apo state (PDB: 1dg3). Amino acids representing the start (K207) and the respective endings (S213, D216, R223) of the replaced regions in $\alpha 4'$ variants are shown. *Right:* structures of helices $\alpha 4$, $\alpha 4'$ and the K-Loop of the respective GBP1 $\alpha 4'$ variants predicted by AlphaFold2. The GGGS-linkers are shown replacing the indicated regions. The structure of wild-type GBP1 of the left panel is superimposed to the AlphaFold2 prediction (transparent).

We have changed **Fig 2B** to better visualize the rationale of the GGGS-linkers in the $\alpha 4'$ variants.

On this point, although the EM suggests the overall architecture is maintained in the mutants, the authors should confirm this for example, are gel filtration profiles similar, so that we are certain the localised effects of $\alpha 4'$ helix mutations are responsible for the changes in oligomerisation efficiency, and not structural changes induced in the protein elsewhere.

To address this concern, we performed analytical size-exclusion chromatography (SEC) and circular dichroism (CD) measurements with the purified $\alpha 4'$ variants, wild-type GBP1, and the GBP1 RK(227-228)EE mutant (Vopel *et al.*, 2010). The RK(227-228)EE mutant fails to build the salt bridge network between helix $\alpha 4'$ and the GED, which locks the GED to helix $\alpha 4'$. Consequently, this mutant exists in an open conformation in which the GED is flipped out. We used this mutant as reference for a GBP1 variant with an altered tertiary protein structure.

As indicated in **Fig R4A**, there are no significant differences in the analytical SEC profiles between the $\alpha 4'$ mutants and the wild-type GBP1. As expected, the RK(227-228)EE mutant eluted earlier in gel filtration compared to wild-type, in line with a more extended conformation. CD measurements indicated no significant secondary structure changes of wild-type GBP1 and the $\alpha 4'$ mutants (**Fig R4B**). Since only the tertiary structure is altered in GBP1 RK(227-228)EE, the spectrum and secondary structure prediction did not differ from wild-type GBP1. Based on these data, there is no hint that the introduced mutations in helix $\alpha 4'$ change the overall secondary or tertiary structure of the protein.

Figure R4. Analytical SEC and circular dichroism of helix $\alpha 4'$ variants. (A) Analytical SEC. Wild-type GBP1 and the open GBP1 mutant RK(227-228)EE (Vopel *et al.*, 2010) were used as references for changes in the overall protein architecture. The peak of the monomeric wild-type GBP1 species is highlighted in yellow for comparison. (B) Circular dichroism. Top: CD spectra normalized according to the cell path length, protein concentration and number of amino acid residues. Bottom: Secondary-structure determination using CDNN 2.1 (Bohm *et al.*, 1992).

We included both the analytical SEC and the CD data as **Appendix Figure S6** and revised the corresponding passage in the result section (**lines 197-199**) and added the CD spectroscopy to the method section.

Although the authors attempt to explain the increased GTPase activity of the 207-213 mutant, there is limited discussion of why $\alpha 4$ mutants cannot carry out the second hydrolysis step. This should be explored further.

Indeed, our GTP hydrolysis assay (Fig 3A-C) indicates that $\alpha 4'$ variants lack the ability to hydrolyze GDP. As shown in the crystal structures (Movie EV3), helix $\alpha 4'$ and $\alpha 3$ are in close proximity to the dimeric interface and contribute to intermolecular contacts in the dimer. We discuss in the manuscript (lines 414-424) that deletions of helix $\alpha 4'$ may affect the interplay with helix $\alpha 3$ and thus, may destabilize the dimer interface in the GDP-bound GBP1 dimer, leading to dissociation.

The GDPase activity of full-length GBP1 is almost vanished when offering GDP instead of GTP (Ince *et al.*, 2021). However, the isolated LG domain lacking the auto-inhibitory GED showed a higher GDPase activity compared to wild-type, due to facilitated GDP-induced dimerization (Ince *et al.*, 2021). To address this issue experimentally and to further validate our hypothesis, we performed analytical size-exclusion chromatography of the isolated LG domains of wild-type GBP1 and helix $\alpha 4'$ variants in the absence of nucleotide (apo) and with GDP-AIF_x and GMP-AIF_x mimicking the first and second hydrolysis step, respectively (Fig R5A). GBP1 LG-wt and the $\alpha 4'$ variants LG- $\Delta 207-213/216/223$ eluted as monomers in the apo state. In presence of GDP-AIF_x, all formed a prominent dimeric species. In presence of GMP-AIF_x, however, only LG-wt formed a dimer, while LG- $\Delta 207-213/216/223$ eluted as monomers. Thus, all $\alpha 4'$ variants were unable to form dimers when mimicking the second hydrolysis step. Furthermore, we determined GDPase activities of the isolated LG domains with GDP as substrate (Fig R5B). For all $\alpha 4'$ variants (LG- $\Delta 207-213/216/223$), the maximum catalytic GDPase activity were drastically reduced as compared to LG-wt, even though nucleotide binding affinities toward mant-GDP resembled that of LG-wt (Fig R6). These results support our hypothesis that dimer dissociation upon the first hydrolysis step is favored over consecutive GDP hydrolysis when helix $\alpha 4'$ is altered.

Figure R5. GDP hydrolysis of GBP1-LG $\alpha 4'$ mutants. (A) Analytical size-exclusion chromatography of isolated LG domains. Dimeric fractions are highlighted in blue, monomeric fractions in yellow. (B) Specific activity of cooperative GDP hydrolysis of isolated LG domains. Initial hydrolysis rates ($n=2$) were normalized to the protein concentration and plotted against protein concentration. The dashed line represents a fit for WT using Eq. 2 ($K_d = 110 \mu\text{M} \pm 50 \mu\text{M}$, $k_{max} = 120 \text{min}^{-1} \pm 40 \text{min}^{-1}$).

Figure R6. Mant-GDP binding to GBP1 LG constructs. Fluorescence of mant-GDP (0.5 μM) at varying GBP1 concentration for indicated isolated LG domain constructs. Data points are averages from three independent experiments and are represented by mean \pm SD. Equilibrium dissociation constants K_d were calculated by fitting a quadratic equation to data.

We have added the data of the size-exclusion runs and GDPase assay to our manuscript as **Fig 3D and 3E**, the data of the nucleotide-binding assay as **Fig EV3B**, and revised the corresponding passage in the result section (**lines 265-280**).

- Could the authors clarify their conclusion at line 324, suggesting the coordinated movement couples nucleotide hydrolysis to oligomerisation. If pivot point mutants can encapsulate *E. coli* efficiently, and oligomerisation isn't affected, how does this support the pivot point facilitating a coordinated movement to allow oligomerisation? The authors should provide evidence for their explanation that LPS of *E. coli* is responsible for stabilising oligomerisation, in a mutant that should have dramatically reduced oligomerisation.

Indeed, we have shown that GTP hydrolysis is affected in the pivot point mutants D199A/K and that GTP-induced polymerization is dramatically reduced (**Fig 4E, F**). Furthermore, also the M139 mutants, in which the coordinated movements of helix $\alpha 3$ - $\alpha 4'$ are hindered either by locking helix $\alpha 3$ to $\alpha 4'$ or by uncoupling helix $\alpha 3$ from $\alpha 4'$, display a similar phenotype (**Fig 4B, C and EV3A**). Together with the analysis of the nucleotide-dependent structural transitions (**Fig 4A, D and Movie EV3**), we now adjust our conclusion to suggest that the pivot point is involved in the coordinated movements in the LG domain upon nucleotide binding and, possibly, nucleotide hydrolysis.

Since these mutants are still able to encapsulate *E. coli* efficiently (**Fig 4G**), we hypothesized that LPS on the bacterial surface stabilizes the oligomeric assembly, as previously described by Kutsch *et al* (2020). We performed additional experiments to provide evidence that LPS is indeed responsible for stabilizing oligomerization. In the light scattering-based polymerization assay, both pivot point mutants show a clear increase in GTP-induced polymerization with increasing LPS concentrations, followed by a decrease, indicating efficient GTP hydrolysis (**Fig R7**). Based on these data, pivot point mutants in absence of LPS do not polymerize efficiently, but LPS present on the bacterial surface facilitates and stabilizes GBP1 oligomerization and also allows oligomerization of these mutants.

Figure R7. Light scattering-based polymerization assay of pivot point mutants in presence of LPS. LPS was added to the reaction at the indicated concentration at constant GBP1 concentration (10 μ M). Polymerization was then induced by GTP.

These data neatly close the gap between the light scattering-based polymerization assay and the bacterial binding assay. We included them as **Fig 4H** in our manuscript and revised the corresponding passage in the result (**lines 344-349**) and discussion section (**lines 433-434**).

Minor points

- In line 146 it's suggested that the MD and GED domains don't contain oligomerisation interfaces. With the map at this resolution, it is hard to rule out weaker contacts between these domains, especially at the GED closer to the membrane. Flexibility within the oligomer may mean these contacts have been averaged out, especially given the LG domains are likely to have more extensive contacts and a more rigid lattice. It is recommended that the authors clarify in the text that contacts in these regions could have been lost through averaging.

The text has been revised as suggested (**lines 153-155**).

- Could the difference between the BPL and Folch Lipids be discussed? Is there an obvious reason membrane-independent oligomerisation is favoured for the Folch lipids?

There is no detailed information about the composition of the two used brain extracts in the literature and provided by the suppliers. Brain polar lipid (BPL) extract is derived from a total lipid extract by precipitation with acetone (<https://avantilipids.com/product/141101>), which is not done in the Folch extraction procedure (Folch *et al*, 1957). Acetone precipitates phospholipids, while, for example, glycolipids and other simple lipids dissolve readily in acetone (Hanahan *et al*, 1951).

To address this question and study the differences between Folch and BPL liposomes, we performed acetone precipitation with the Folch extract according to Hanahan *et al*. (1951). However, there was no significant improvement in liposome-binding with lipids derived from this procedure (**Fig R8**). Next, we probed whether specific lipids, if supplied to Folch liposomes, may rescue GBP1-binding. Since GBP1 has a C-terminal polybasic motif (⁵⁸⁴RRR⁵⁸⁶), we initially tested lipids with negatively charged head groups (10% DOPS or 10% PI(4,5)P₂) in our FRET-based liposome binding assay, which, however, did not increase GBP1 binding (**Fig R8**). Finally, we supplemented Folch lipids with 25% cholesterol in order to modulate membrane fluidity and elasticity. Indeed, this approach increased membrane-binding in the Folch mixture almost to the same level as seen for BPL liposomes. Thus, high membrane fluidity and elasticity seem to promote GBP1 membrane-binding, but we cannot ascertain whether and if, how such a difference is achieved between Folch and BPL liposomes.

Figure R8. FRET-based liposome binding assay. Ratiometric FRET efficiencies of GBP1-Q577C-AF488 (donor) incubated with indicated liposomes supplemented with Liss Rhod PE (acceptor) and indicated nucleotides. Data from three independent replicates are shown as mean \pm SD.

We included these data as **Appendix Figure S1B** and revised the corresponding passage in the result section (**lines 138-142**). In addition, we now discuss why membrane-independent oligomerization appears to be favored for Folch liposomes versus BPL (**Fig 1D**), which may merely reflect the consumption of soluble oligomers at the expense of coatomers in the presence of BPL liposomes (**lines 363-366**). This is supported by the observation of rare binding events of soluble polymers to coated BPL liposomes (**Fig R9**). We included Fig R9 as **Appendix Figure 2** and revised the corresponding passage in the result section (**lines 134-136**).

Figure R9. Rare binding events of soluble polymers to coated BPL liposomes. Oligomerization initiated with GDP-AIFx. Arrows: attached polymers. Arrow heads: soluble polymers/disks.

- The cross-linked oligomer band for the 207-223 mutant in Figure 2C is described as "completely abolished". However, it appears there is still a faint band at the top of the gel. The authors should acknowledge this.

Thank you for pointing this out. This has been revised as suggested.

- Although not in the scope of this manuscript, this work would be greatly strengthened by tomograms of polymers bound to *E. coli*. This would confirm the conformation they are proposing may have a more physiological role.

Indeed, these are exciting, but highly demanding experiments requiring sample optimization and data collection under S2 conditions, which is not available to us. Furthermore, the suggested experiments have already been described some time ago in a bioRxiv manuscript by Zhu *et al* (2021) (referenced in our manuscript). We now more specifically refer to this study in our discussion (**line 375**).

Referee #3:

In this manuscript, Weismehl and colleagues explored the molecular mechanism underlying the oligomerization process of human GBP1. Combining cryo-EM, cryo-ET, and many biochemical approaches, the authors convincingly identified critical interactions between the $\alpha 3$ and $\alpha 4'$ helices of human GBP1 that regulate its oligomeric state. This is a beautiful piece of work. I have little to criticize and fully support publication on EMBO J.

Thank you very much for the positive evaluation!

Minor concerns:

1. What is the lipid-specificity of GBP1? The authors showed in Fig. 1D that GBP1 preferentially forms lipid-independent oligomers in the presence of Folch lipids, while coats BPL. What causes this difference? Is it related to the similarity with bacterial membrane?

Concerning the difference of GBP1-binding to Folch and BPL liposomes, we would like to refer the referee to our response to referee #2, minor point, comment 2.

Regarding the difference to bacterial membranes: The major difference appears to be the presence of LPS in bacterial membranes which favor GBP1 oligomerization. In line with this suggestion, we have now included new data in the manuscript showing that LPS can stimulate GTP-induced oligomerization of pivot point mutants which otherwise show reduced oligomerization capacity (**new Fig 4H**). See also our response to the final major comment of referee 2.

2. In Fig. 2A, it would be more helpful to label the shift of the $\alpha 4'$ helix in a more obvious way - it took me a while to identify the mentioned shift.

Thanks for pointing this out. We have updated **Fig 2A** for better clarity.

3. In Fig. 2B, does the length of the GS-linker match the length of the substituted residues, especially for d207-216 and d207-223 mutants?

We modified **Fig 2B** to better explain the rationale in designing the mutants. See also major point 2 of referee #2.

Bohm G, Muhr R, Jaenicke R (1992) Quantitative analysis of protein far UV circular dichroism spectra by neural networks. *Protein Eng* 5: 191-195

Folch J, Lees M, Sloane Stanley GH (1957) A simple method for the isolation and purification of total lipides from animal tissues. *J Biol Chem* 226: 497-509

Hanahan DJ, Turner MB, Jayko ME (1951) The isolation of egg phosphatidyl choline by an adsorption column technique. *J Biol Chem* 192: 623-628

Ince S, Zhang P, Kutsch M, Krenczyk O, Shydlovskiy S, Herrmann C (2021) Catalytic activity of human guanylate-binding protein 1 coupled to the release of structural restraints imposed by the C-terminal domain. *FEBS J* 288: 582-599

Ji C, Du S, Li P, Zhu Q, Yang X, Long C, Yu J, Shao F, Xiao J (2019) Structural mechanism for guanylate-binding proteins (GBPs) targeting by the *Shigella* E3 ligase IpaH9.8. *PLoS Pathog* 15: e1007876

Jumper J, Evans R, Pritzel A, Green T, Figurnov M, Ronneberger O, Tunyasuvunakool K, Bates R, Zidek A, Potapenko A et al (2021) Highly accurate protein structure prediction with AlphaFold. *Nature* 596: 583-589

Kutsch M, Sistemich L, Lesser CF, Goldberg MB, Herrmann C, Coers J (2020) Direct binding of polymeric GBP1 to LPS disrupts bacterial cell envelope functions. *EMBO J* 39: e104926

Mirdita M, Schutze K, Moriwaki Y, Heo L, Ovchinnikov S, Steinegger M (2022) ColabFold: making protein folding accessible to all. *Nat Methods* 19: 679-682

Vopel T, Syguda A, Britzen-Laurent N, Kunzelmann S, Ludemann MB, Dovengerds C, Sturzl M, Herrmann C (2010) Mechanism of GTPase-activity-induced self-assembly of human guanylate binding protein 1. *J Mol Biol* 400: 63-70

Zhu S, Bradfield CJ, Mamińska A, Park E-S, Kim B-H, Kumar P, Huang S, Zhang Y, Bewersdorf J, MacMicking JD (2021) Cryo-ET of a human GBP coatomer governing cell-autonomous innate immunity to infection. *bioRxiv*: 2021.2008.2026.457804

Dear Oliver,

Thank you for the submission of your revised manuscript to The EMBO Journal. We have now received the comments of the three referees that were asked to re-evaluate your study (included below). As you will see, all referees are satisfied with the revision, acknowledge that their previous concerns have been addressed thoroughly and satisfactorily, and support publication of the study.

From the editorial side, there are a few minor changes that we need from you before we can proceed with acceptance of the manuscript:

- Please change "Material and Methods" to "Materials and Methods".
- Please add the heading "Methods and Protocols" after your "Reagents and Tools Table", in the Materials and Methods.
- Please change the heading of your "Conflict of interests" statement to "Disclosure and competing interests statement".
- Please remove the movie legends from the manuscript file; instead, each movie file should be zipped together with its legend in a Word/text file.
- Please note that the final dimensions of the synopsis image are 550 pixels (width) x 300-600 pixels (height can be variable within this range). Could you please resize your synopsis image accordingly?
- As soon as these issues are resolved, I might contact you again to discuss with you a few suggestions for minor improvements in the title, abstract and synopsis text.

Please also note that as part of the EMBO publications' Transparent Editorial Process, The EMBO Journal publishes online a Peer Review File along with each accepted manuscript. This File will be published in conjunction with your paper and will include the referee reports, your point-by-point response and all pertinent correspondence relating to the manuscript. You can opt out of this by letting the editorial office know (contact@embojournal.org). If you do opt out, the Peer Review File link will point to the following statement: "No Peer Review File is available with this article, as the authors have chosen not to make the review process public in this case."

We look forward to seeing a final version of your manuscript as soon as possible. Please use this link to submit your revision: <https://emboj.msubmit.net/cgi-bin/main.plex>

Best regards,

Ioannis

Referee #1:

I am satisfied the authors have answered my own queries. I have also read the full review and point-by-point response and believe that they have answers all of the reviewer questions sufficiently for publication.

Referee #2:

We thank the authors very much for addressing all of our concerns so thoroughly. Great work! This is an exciting high quality study and look forward to its publication.

Referee #3:

The authors have addressed my questions satisfactorily. I support publication as is.

All editorial and formatting issues were resolved by the authors.

Dear Oliver,

I am very pleased to inform you that your manuscript has been accepted for publication in The EMBO Journal.

Best regards,

Ioannis
